# Polarity is all you need to learn and transfer faster

## Abstract

Natural intelligences (NIs) thrive in a dynamic world – they learn quickly, sometimes with only a few samples. In contrast, Artificial intelligence (AI) has achieved supra (-human) level performance in certain AI settings, typically dependent on a prohibitive amount of training samples and computational power. What design principle difference between NI and AI could contribute to such a discrepancy? Here, we propose an angle based on a simple observation from NIs: post-development, neuronal connections in the brain rarely see polarity switch. We demonstrate with simulations that if weight polarities are adequately set *a priori*, then networks learn with less time and data. We extend such findings onto image classification tasks and demonstrate that fixed polarity, not weight, is a more effective medium for knowledge transfer between networks. We also explicitly illustrate situations in which *a priori* setting the weight polarities is disadvantageous for networks. Our work illustrates the value of weight polarities from the perspective of statistical and computational efficiency during learning.

## 1 Introduction

Natural intelligences (NIs), including animals and humans, thrive in a dynamic world. Often, NIs learn quickly with just a few samples. Artificial intelligences (AIs), specifically deep neural networks (DNNs), can now compete with or even surpass humans in certain tasks, e.g., GO playing (Silver et al., 2017), object recognition (Russakovsky et al., 2015), protein folding analysis (Jumper et al., 2021), etc. However, DNN is only capable of achieving such when a prohibitive amount of data and training resources are available. Such a gap on learning speed and data efficiency between NI and AI has baffled and motivated many AI researchers. A subfield of AI is dedicated to achieving few-shot learning using DNNs (Hoffer & Ailon, 2015; van der Spoel et al., 2015; Vinyals et al., 2016; Snell et al., 2017; Finn et al., 2017). Many research teams have achieved amazing performances on benchmark datasets (Lazarou et al., 2022; Bendou et al., 2022). However, the products of such engineering efforts greatly deviate from the brain. What are the design principle differences between NIs and AIs that contribute to such a learning efficiency gap? In this paper, we propose one possiblility of such a design difference - we could move AI one step closer to NI-level learning efficiency by applying just one simple design principle from NI.

NIs are blessed with hundreds of millions of years of optimization through evolution. Through trial and error, the most survival-advantageous circuit configurations emerge, refine, and slowly come into the form that can thrive in an ever-changing world. Such circuit configurations get embedded into our genetic code, establishing a blueprint to be carried out by development. Among the many configurations, circuit rules, and principles that formed through evolution, one theme stands out, one that neuroscientists celebrate and yet is overlooked by the machine learning community: post-development, neuronal connections in the brain rarely see polarity switch (Spitzer, 2017). After development, NIs learn and adapt through synaptic plasticity – a connection between a pair of neurons can change its strength but rarely its excitatory or inhibitory nature; on the contrary, a connection (weight) between a pair of units in a DNN can freely change its sign (polarity). In fact, polarity change in the adult brain is hypothesized to be associated with depression, schizophrenia, and other illnesses (Spitzer, 2017). For the rare times such phenomenon have been observed, they never appeared in sensory and motor cortices (Spitzer, 2017) where visual, auditory and motor processing take place. It seems a rather rigid design choice to fix a network's connection polarity. We wonder why the biological networks settled into such a learning strategy: Is it a mere outcome of an implementation-

level constraint? It could be just hard for synapses to switch polarities. Or could it be that NIs found out polarity pattern is an effective medium for transferring knowledge across generations?

This paper provides some thoughts and evidence relevant to these questions. We first investigate why brains pre-set neuronal connection polarities by assessing what we *gain* through setting weight polarity *a priori* (Sec 2-4). We discuss in theory the trade-off between representation capacity and learning speed when weight polarity is fixed for networks. We then propose an SGD-based polarity-fixed learning algorithm: **Freeze-SGD**. We experimentally show that if the weight polarities are *adequately* set *a priori*, then networks can learn with less time and data (simulated task (Sec 2) + two image classification tasks (Sec 3)). We call a network with fixed polarity **Frozen-Net**, and we discuss how the quality of the polarity configuration affects a DNN's learning efficiency (**Sufficient-Polarity** vs. **RAND-Polarity**). We further find transferring and fixing polarities is even superior to transferring weights (Sec 4). Our results point to an unexplored direction in the machine learning community: polarity, not weight, may be the more effective and compressed medium for transferring knowledge between networks. To complete our discussion, we further discuss what we *lose* when weight polarities are set *a priori* (Sec 5). Frozen-Nets have reduced representation capacity; therefore, a randomly configured network may not even have the capacity to represent a simple problem such as XOR (Def 2.2). We theoretically prove and experimentally show that if the polarities are set randomly, the probability of a single-hidden-layer network learning XOR increases exponentially as a function of its size (i.e., number of hidden units). We also experimentally show that a sufficiently sized network, even when its polarities are randomly picked, can learn with less time and data than an equally sized network without fixed polarities.

By discussing both the advantages and disadvantages of fixing weight polarity, we provide some insights on how to make AI more statistically and computationally efficient; we also show polarity pattern is an effective medium for transferring knowledge.

## 2 WHAT DO WE GAIN BY SETTING WEIGHT POLARITY *a priori*?

Networks need both positive and negative weights to funcion (Wang et al., 2022) - a DNN with all non-negative weights is not a universal approximator. Constraining a network's weight polarity pattern limits its representation capacity: when only half of the range is available to each connection, the reduction in total possible network patterns grows exponentially with more edges in the network. It seems counter-intuitive for any network to have willingly chosen to give up on a vast portion of its representation capacity. Are they gaining elsewhere? Our thought is: maybe they learn faster. Below we provide theoretical discussions and experimental evidence.

**Definition 2.1** (DNN). *Let $W^{(l)} \in \mathbb{R}^{n_l \times n_{l-1}}$ be the input weights of layer $l$, $n_l$ be the number of units in layer $l$, $b^{(l)} \in \mathbb{R}^{n_l}$ be the bias terms of layer $l$, for $l \in \{1, \dots, L\}$. $l = 0$ is the input layer, $n_0 = dim(x)$. Let layer $l$ be $\sigma^{(l)}(x) = \sigma(W^{(l)}x + b^{(l)})$.*

$$P(x) = \sigma^{(L)} \circ \sigma^{(L-1)} \circ \dots \circ \sigma^{(1)}(x), P(x) \in \mathbb{R}^{n_L};$$
$$F(x) = \text{softmax}(P(x))$$

*DNN is $F(x)$. For the non-linearity $\sigma$, we define it to be ReLU throughout the paper.*

$$\sigma(x_j) = \begin{cases} 0 & x_j < 0 \\ x_j & x_j \geqslant 0 \end{cases} \qquad \sigma(x) = (\sigma(x_1), \dots, \sigma(x_{n_l}))$$

**Lemma 2.1** (capacity-speed trade-off). *If the weight polarities are set a priori, such that the function is still representable, then the network can learn faster.*

We prove Lemma 2.1 for single-hidden-layer networks, with the following assumptions:

> Assumption 1: The weights take on discrete values. This is essentially true for all DNNs implemented on silicon chips where all continuous variables are discretized;
>
> Assumption 2: Exhaustive search as the learning algorithm.

See proof on page 20.

The theory, albeit proved under constrained settings, argues for a trade-off between network representation capacity and learning speed. Next, we test with simulation to show that networks

learn more quickly when polarities are set *a priori* in such a way that the function is still representable (Figure 1 panel B). With input space $\mathcal{X}$, a function $f$ is representable by a DNN $F$ when $\forall x \in \mathcal{X}, \epsilon > 0, \ |f(x) - F(x)| < \epsilon$.

We design our freeze training procedure to be exactly the same as SGD (Adam optimizer) except for one single step: after each batch, all weights are compared to the preset polarity template, and if any of the weight has switched polarity in the last batch, it is reverted back to the desired sign (see algorithm 1). As our goal is to see the pure effect of fixing weight polarity, we did not adopt any bio-plausible learning algorithms as they may introduce confounding factors.

We compared four training procedures in general:

1. **Frozen-Net sufficient-Polarity**: Weight polarities were set *a priori*. The polarity pattern was chosen based on a rule that ensures the configuration is adequate for learning the task, i.e., the polarity pattern carries expert knowledge about the data.

2. **Frozen-Net RAND-Polarity**: Weight polarities were set *a priori* randomly $Bernoulli(0.5)$. The polarity pattern does not carry any information about the data.

3. **Fluid-Net RAND-Polarity**: Weights (containing polarities) were initialized randomly; weight polarities were free to change throughout the training procedure.

4. **Fluid-Net sufficient-Polarity** Weight polarities (not magnitudes) were initialized with prior knowledge; weight polarities were free to change throughout the training procedure. This scenario will only be discussed in the next section 3 on image classification tasks.

We used 5-dimensional XOR (XOR-5D) as our simulation binary classification task; only the first two dimensions are relevant to the task, and the remaining three dimensions are noise following a normal distribution $\mathcal{N}(1,1)$ (Figure 1 panel A). For Frozen-Net sufficient-Polarity, the polarity template is pre-set in this way: for each hidden unit, the polarity of the output edge is the sign product of the first two dimensions' input weights.

**Definition 2.2** (XOR). *Let $x \in \mathbb{R}^2$ and $t_1, t_2 \in \mathbb{R}$,*

$$f(x) = \begin{cases} 0 & [x_1 \geq t_1 \text{ and } x_2 \geq t_2] \quad \text{or} \quad [x_1 < t_1 \text{ and } x_2 < t_2] \\ 1 & [x_1 \geq t_1 \text{ and } x_2 < t_2] \quad \text{or} \quad [x_1 < t_1 \text{ and } x_2 \geq t_2]. \end{cases} \tag{1}$$

To have the best controlled experiments, we fixed the magnitude distribution, architecture, training samples (n varies, see figure), batch sequence of the training data, and the validation samples (n=1000) across all scenarios. More details about the experiments can be found in the appendix Sec C. We first tried four different weight reset methods (details in Algo 1) and found they give us similar results and thus did not matter for our primary questions (supplementary figure B.1). For the rest of this paper, we chose the *posRand* method whereby we set weights to a small random number of the correct sign.

When the polarities are fixed in such a way that it is sufficient to learn the task (red), networks always learn faster than networks without polarity constrains (blue) (Fig 1 panel B). This advantage is true across all data scarcity levels and is particularly valuable when data is scarce. When only 60 or 72 training samples were available, Frozen-Net sufficient-polarity on average take 58% and 48% of the standard Fluid-Net training time respectively to reach the same level of accuracy. When weight polarities are randomly chosen (green), networks learn as fast as their Fluid-Net counterparts (blue), and sometimes faster (e.g., training samples = 80 and 92).

Frozen-Net sufficient-Polarity not only saves time, but also saves data (Fig 1 panel C). At convergence, Frozen-Net sufficient-Polarity (red) takes fewer samples to reach the same level of accuracy when compared to the standard Fluid-Net (blue). Randomly configured Frozen-Net (green) uses similar and oftentimes less data than Fluid-Net (blue) (e.g., to reach $80\%$ accuracy, Frozen-Net RAND-polarity uses less data than Fluid-Net).

We showed that setting weight polarity *a priori* makes networks learn in less time, with less data, provided the function is still representable. Even randomly configured Frozen-Nets show comparable and sometimes better performance than Fluid-Nets. Such a result is striking from an optimization perspective: Frozen-Net is at a disadvantage by design because the weight resetting step in Freeze-SGD 1 fights against the gradient update - part of the error update information is lost in this process. Regardless of this disadvantage during optimization, our Frozen-Net sufficient-polarity consistently

outperforms Fluid-Net; even Frozen-Net RAND-Polarity is never worse than Fluid-Net. Combined, these results show we may help AIs to learn quickly and with fewer samples by doing two things: 1) fix weight polarities, and 2) choose polarity pattern wisely. We will tease apart the effect of these two factors in the next section.

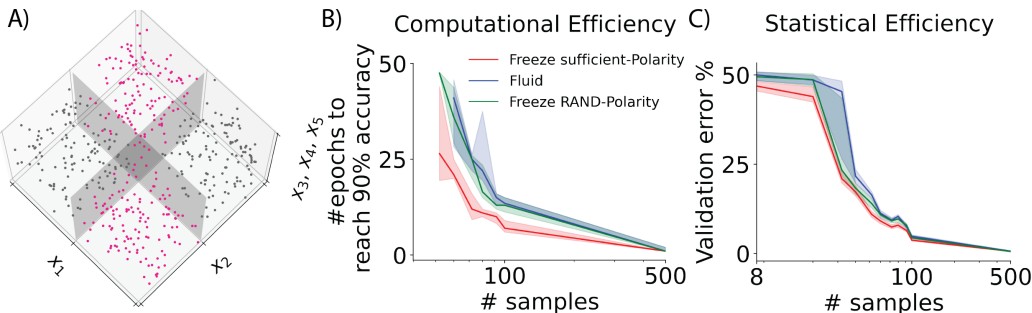

Figure 1: **Adequately setting weight polarity *a priori* makes networks learn more quickly (fewer epochs) and with less data (fewer training samples).** Single hidden layer networks with 64 hidden units are trained to learn XOR-5D. In all scenarios, networks were trained for 100 epochs. A) XOR-5D, a binary classification problem where only the first two dimensions are relevant to the task (XOR), and the other three dimensions are noise following a normal distribution $\mathcal{N}(1,1)$. B) Computational efficiency: Setting weight polarity in a data-informed way (red) makes networks learn more quickly – this is true across all data-scarcity levels; when weight polarities are randomly fixed (green), networks learn as fast as their fluid counterparts (blue), and sometimes faster (e.g., training sample sizes = 80 and 92). C) Statistical efficiency: At convergence, adequately configured freeze networks (red) achieve the same level of performance (% error) with less data; randomly configured freeze networks (green) use similar and oftentimes less data than fluid (blue), e.g., to reach 80% accuracy (20% error). For all experiments, n=50 trials. All curves correspond to medians with shaded regions 25th-75th percentiles.

## 3 EFFECTIVENESS OF FROZEN-NET IN IMAGE CLASSIFICATION TASKS

To prove the effectiveness of our Frozen-Net strategy for more complex tasks, we extended the experiments in Fig 1 to image classification tasks (Fig 2). Such complex learning tasks do not have simple and explicit polarity configuration rules as in XOR-5D. We exploited an alternative strategy by using ImageNet trained polarities (IN-Polarity). A Frozen-Net IN-Polarity has its weight *magnitudes* initialized randomly, its weight *polarities* initialized and fixed to match the IN-polarity pattern. We also tested Fluid IN-Polarity where networks were initialized to the exact same pattern as Frozen-Net IN-Polarity, except polarities are free to switch in this scenario. This pair of comparison help us to understand which of the two factors contributed more to the performance gain: fixing polarities or knowledge transfer through polarity pattern. We trained and tested networks on the Fashion-MNIST (grayscale) and CIFAR-10 (RGB-color) datasets, using AlexNet network architecture (Krizhevsky et al., 2017). For both datasets, we trained for 100 epochs, with lr=0.001. The AlexNet IN-weights were obtained here. Worth emphasizing, we controlled all conditions to follow the same weight magnitude distribution at initialization (supp Fig B.6) Specifically, we randomly initialized the networks following normal procedures: Glorot Normal for conv layers, Glorot Uniform for fc layers; then either fixed the polarities as is (RAND-Polarity) or flipped the polarities according to IN template (IN-Polarity), introducing no change to the magnitude distributions.

Across the board, Frozen-Nets IN-polarity (red) always learn with fewer samples than Fluid-Net (blue) (Fig 2 first column). When only 100 training images were available (across the 10 classes), Frozen-Net IN-Polarity (red) yields 7% less validation error at convergence compared to Fluid-Net (blue) in the Fashion-MNIST task and 9.4% less error for CIFAR-10. Such a gain is mostly brought by knowledge transferred through polarity pattern (pink vs. blue, 6% gain for Fashion-MNIST; 8.4% gain for CIFAR-10). Fixing the polarities can further bring performance gain: for CIFAR-10, 1% gain at 100 training samples, and up to 3% at 50000 training samples. The gain from Fluid IN-Polarity to

Freeze-Net IN-Polarity can be statistically significant (Fig 3, pink line). When the polarities are fixed randomly (RAND-polarity, green), networks never perform worse than Fluid-Net (blue).

Figure 2: **DNNs with frozen IN-Polarity learn more quickly and with less data in image classification tasks.** A) Experiments on Fashion-MNIST image classification dataset. From left to right: 1) Statistical efficiency: Frozen-Nets with IN-polarity (red) always learn with fewer samples than Fluid-Net (blue); the majority of the gain is contributed by the knowledge transferred from initial polarity configuration (pink vs. blue); Frozen-Nets RAND-Polarity (green) never perform worse than Fluid-Net (blue). 2) Frozen-Net IN-Polarity always have a higher chance of reaching $80\%$ validation accuracy than Fluid-Net; Frozen-Net RAND-Polarity have comparable and sometimes a higher chance of reaching $80\%$ validation accuracy than Fluid-Net. 3) Computational efficiency: Frozen-Net IN-Polarity always take less time than Fluid-Net to reach $80\%$ validation accuracy; again, effective knowledge transfer from preset polarity pattern is the major contributing factor (pink vs. blue);Frozen-Net RAND-Polarity takes a similar number of computational iterations as Fluid-Net. B) Same as A except experiments were on CIFAR-10 dataset and the validation accuracy threshold is at $50\%$. Gray lines in the first column correspond to the validation accuracy thresholds used to plot the next two columns. For a comprehensive view of performances at different thresholds, see Supp Fig B.2. For statistical significance of the difference, see Fig 3. Both datasets: n=20 trials, 100 epochs, lr=0.001. No data augmentation was performed.

Across the board, Frozen-Net IN-Polarity always learns with less time than Fluid-Net (Fig 2 third column). Majority of such a gain is brought by polarity pattern knowledge transfer. Frozen-Net RAND-Polarity takes a comparable number of computational iterations as Fluid-Net. Furthermore, not every network is able to reach the specified accuracy threshold (Fashion-MNIST: $80\%$, CIFAR-10: $50\%$; Fig 2 second column). Across the board, Frozen-Net IN-Polarity has a higher chance of passing the specified accuracy threshold than Fluid-Net; Frozen-Net RAND-Polarity has an equal and sometimes higher chance than Fluid-Net (Fashion-MNIST 1000 & 2000 samples; CIFAR-10 10000 & 25000 samples). These observations are true across different validation accuracy thresholds (Supp Fig B.2).

Worth noting, IN-Polarity initialized networks in general shows more consistent performance across trials, compared to the Fluid setting, for both Frozen-Net and Fluid-Net. This is especially obvious for the statistical efficiency plots: across the board, IN-Polarity always shows less variation in its performance on validation error (shaded area marks $25^{th}$-$75^{th}$ percentiles).

To provide a lens into the dynamic of polarity switch throughout learning, we analyzed the ratio of weight parameters (excluding bias terms) that had polarity switch between two epochs for Fluid RAND-Polarity for the first 50 epochs - indeed, there are more polarity switch early on in training

and the ratio decays throughout training. Such a trend is true across layers and across training sample sizes. These suggest polarities are mostly learnt early on during training but also remain dynamic throughout the learning process.

In sum, we are able to extend our simulation results in Sec 2 to more complex image classification tasks: Frozen-Net IN-polarity consistently learns with less data and time and does so with a higher probability of success compared to Fluid-Net; the majority of the performance gain is brought by knowledge embedded in the initialized polarity pattern, with further gain possible by fixing weight polarities; Frozen-Net RAND-polarity perform as well as Fluid-Net, sometimes better.

## 4 TRANSFERRING AND FIXING POLARITY IS SUPERIOR TO TRANSFERRING WEIGHTS

From a transfer learning perspective, our Frozen-Net strategy essentially transfers weight polarities instead of weights per se. How does polarity transfer compare to the traditional finetune (weight transfer) strategy? This time, instead of randomly initializing the weight magnitudes, we initialized the weight magnitudes based on the ImageNet-trained weights (IN-Weight). We compared them with Frozen-Net IN-Polarity by plotting their differences ($\Delta$={LEGEND} - {Freeze IN-Polarity}) in Fig 3 (the original curves before taking the differences can be found in Supp Fig B.3).

The orange curves compare weight transfer (Fluid-Net IN-Weight) with our polarity transfer strategy (Frozen-Net IN-Polarity). Across almost all data scarcity levels, our polarity transfer strategy achieves higher accuracy than finetune (first column, orange$\geqslant$ 0). Such a superiority is statistically significant

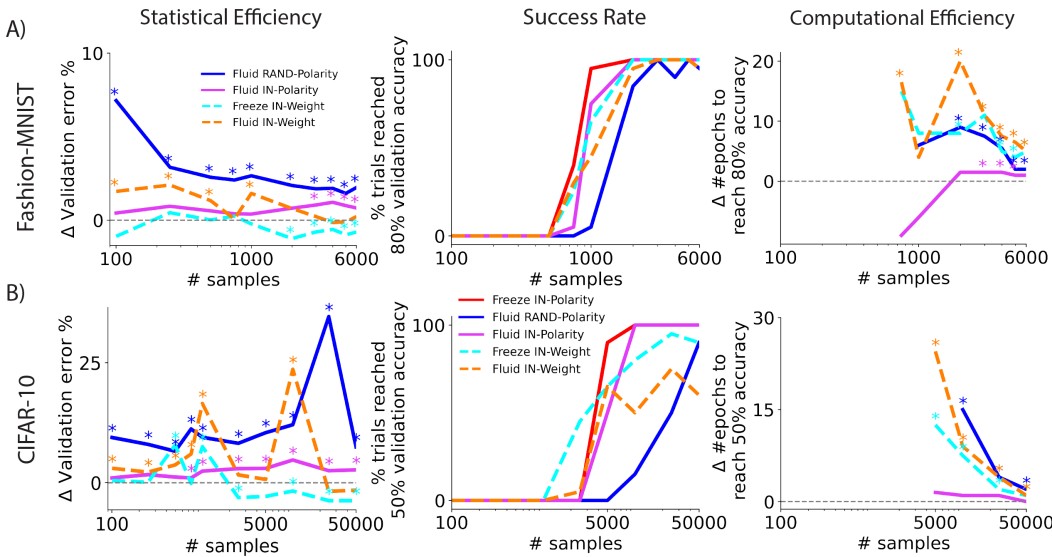

Figure 3: **Across all sample sizes, Frozen-Net with ImageNet-polarities (IN-polarities) learns more quickly than Fluid-Net with ImageNet-weight (IN-weight) initialization.** In first and third columns, curves are the **median difference $\Delta$={LEGEND} - {Freeze IN-Polarity}** (the second term is the red curve in Fig 2). Asterisks (*) indicate statistical significance with Mann-Whitney U two-tail test ($\alpha$=0.05). 1) First column: Transferring and fixing polarity is more effective than transferring weights in terms of data efficiency. This is indicated by orange curve above zero across most sample sizes, and it is statistically significant for small sample sizes. Note the zero line here means LEGEND = Freeze IN-Polarity. 2) Second column: Frozen-Net IN-Polarity has a higher chance of reaching 80% (Fashion-MNIST) / 50% (CIFAR-10) validation accuracy compared to weight transfer (orange curve); 3) Third column: Frozen-Net IN-Polarity always takes less epochs to reach high validation accuracy than weight transfer (orange curve). Both datasets: n=20 trials, 100 epochs, lr=0.001. For the right two columns, the validation accuracy thresholds are the same as in Fig 2

(Mann-Whitney U two-tail) when training data is limited (Fashion-MNIST: $\leqslant 2000$ samples except 750, CIFAR-10: $\leqslant 1000$ samples). Polarity transfer also allows the networks to learn with higher probability (second column) and fewer epochs (third column, curve $\geqslant 0$). Such faster learning is true regardless of the performance threshold value (Supp Fig B.4). In sum, transferring and fixing polarity is almost always superior to weight transfer both in terms of statistical efficiency and computational efficiency. Such an observation suggests polarity configuration is an effective medium, if not superior to weight pattern, for transferring knowledge between networks.

When networks were transferred with polarities but not frozen (Fluid IN-Polarity pink), they almost always perform better than Fluid IN-Weight (orange, Fig 3 & Supp Fig B.3 to see variance), this is true except rare cases (e.g. CIFAR-10 2500&5000 samples) where the difference is not significant due to the wide performance variation of Fluid IN-Weight.

Surprisingly, when we initialized the Frozen-Net with IN-weight (cyan), there is some gain in performance, but to a limited extent; in fact, it can sometimes be worse. When training data is limited (Fashion-MNIST$\leqslant 1000$, CIFAR-10$\leqslant 1000$), Frozen-Net IN-weight gained little performance (cyan$\approx 0$) and could be detrimental at times (CIFAR-10 500&1000 samples). When training data is more abundant, there is a more consistent accuracy gain by initializing Frozen-Net with IN-weight (Fashion-MNIST gain $\sim 1\%$, CIFAR-10 gain $\sim 4\%$). Such a gain is discounted by more training iterations (third column), and sometimes less likelihood of reaching a high level of performance (second column).

Furthermore, similar to random initialization, weight transfer tend to have wider performance variations compared to polarity transfer, for both Frozen and Fluid networks (Fig B.3). The exact reason behind this observation remains to be explored, our current hypothesis is the stochasticity of sample batching: by only transferring polarity while initializing the magnitudes randomly, the learning process is more robust against such stochasticity. The difference in initialized magnitude distribution between polarity transfer vs. weight transfer could also be a potential contributing factor (supp Fig B.6).

In sum, polarity transfer is superior to weight transfer in most of the scenarios we tested and such a superiority is further secured by fixing the polarities throughout training. To a large extent, weight polarity alone, not weight per se, is enough to transfer knowledge between networks. Giving the additional magnitude information to Frozen-Net can give some performance gain, but only when data and time is abundant; in all other scenarios (i.e., data-limited or time-limited), initializing Frozen-Net with stereotypical weight magnitudes could be detrimental to the learning performance.

## 5    WHAT DO WE LOSE BY SETTING WEIGHT POLARITY *a priori*?

Intelligent agents have limited resources in 1) data, 2) time, 3) space (number of hidden units or other network size parameters), and 4) power ($\propto$time$\times$space, number of flops). In Sec 2,3,4, we showed that fixing weight polarity helps to save on two of these resources, time and data, but with a condition – the polarity configuration must be adequately set such that the network can still represent the function. With fixed polarity, certain configurations will result in networks never being able to learn the task. What is the probability of such unfortunate events happening? We investigate this direction with our simulated task XOR-5D.

Assuming unlimited resources (i.e., perfect learning algorithm, unlimited data and time), we deduced the theoretical probability limit for a single-hidden-layer network to be able to represent XOR (and its high-dimensional variants with task-irrelevant higher dimensions) as a function of the number of hidden units (Supp Theorem D.1). The theoretical results are plotted in Fig 4 panel A (top). In theory, it takes at least 3 units for Fluid-Net and Frozen-Net sufficient-Polarity to be able to learn XOR-5D on every single trial. Such a probability will never be 100% for Frozen-Net RAND-Polarity, no matter how large the networks are. Luckily, the probability grows exponentially with the network size: having 15 hidden units is already sufficient for a randomly configured Frozen-Net to learn XOR with $> 99\%$ probability.

This is exactly what we see in our simulation results (Fig 4 panel A bottom). For a generous amount of data (500 training samples) and learning time (100 epochs), the Frozen-Net RAND-Polarity curve nicely matches the theoretical curve - networks with more than 15 hidden units learn XOR-5D with very high probability, even though their polarities are fixed randomly.

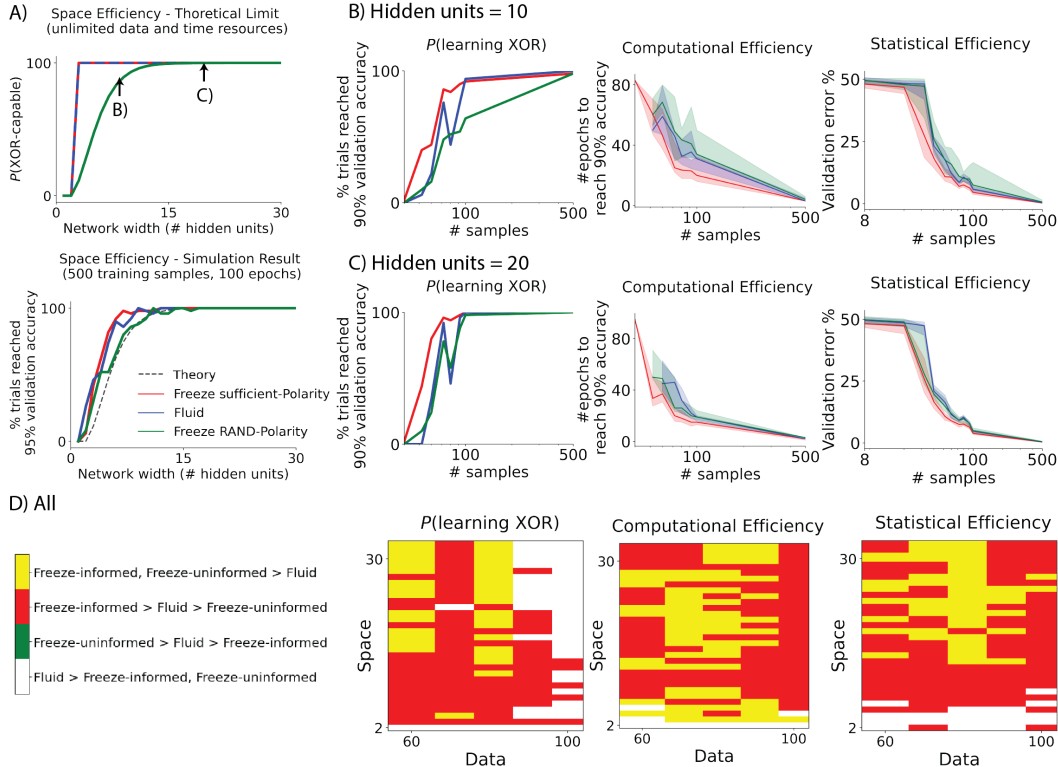

Figure 4: **A network has to be sufficiently large for it to learn more quickly, with less data, if its weight polarity is configured randomly.** A) For a randomly configured Frozen-Net to learn XOR-5D, it has to be sufficiently large. In both the theoretical limit and simulation results, it takes at least 15 hidden units for a Frozen-Net RAND-Polarity (green) to have $> 99\%$ chance of learning XOR-5D; it takes three hidden units, in theory, for an adequately configured Frozen-Net (red) and Fluid-Net (blue) to learn XOR-5D. B) When Frozen-Net RAND-Polarity is not large enough, it has a much lower chance of learning XOR-5D, and when it does learn, it uses more time and data. C) When Frozen-Net RAND-Polarity is sufficiently large, it at least shows the same level of performance as standard Fluid-Nets. D) Frozen-Net sufficient-Polarity shows an advantage over Fluid-Net (red + yellow) in almost all training sample sizes (especially when data is limited) and across all network sizes. Randomly configured Frozen-Nets have an advantage over Fluid-Nets (green + yellow), mostly when the network is sufficiently sized ($> 15$ hidden units). We ran 50 trials for all experiments. All curves represent medians with shaded regions denoting 25th-75th percentiles.

Both theory and simulation results show that we will lose all advantages if weight polarities are fixed but not configured adequately; an example of such is a small, randomly configured Frozen-Net (e.g., 10 hidden units, Fig 4 panel B). Notice that for the same network size, if we ensure the configuration is adequately set (red), then the network learns quickly and is data-efficient. By allowing more space (network size), Frozen-Net RAND-Polarity starts to pick up the advantages in time and data efficiency (Fig 4 panel C). In summary, we gain from setting weight polarity *a priori* only if the polarity is configured adequately (Fig 4 panel D); the adequacy can either be made more probable by having a larger network or can be guaranteed by an explicit configuration rule (e.g., through development for NIs, or explicit rules in our simulations, or transferred polarities from previously trained networks).

## 6 DISCUSSION

We showed in this paper that 1) if the weight polarities are adequately set *a priori*, then networks can learn with less data, time, space, and power; 2) polarity may be a more effective and compressed medium of network knowledge transfer; and 3) while we lose all advantages when the polarities are

fixed but not set adequately, we can regain these advantages by increasing the size of the networks. Below we discuss the novelty of our work and some future directions.

In the literature of bio-plausible artificial neural networks (ANNs), the most related work is on Dale's principle: a single unit's output weights are exclusively excitatory or inhibitory. An exciting attempt of applying such a principle to ANNs achieved performance comparable to multi-layer perceptrons (gray-scale image classification tasks) and VGG16 (CIFAR-10) (Cornford et al., 2021). Our approach differs in several ways; the most fundamental one is ours does *not* require exclusivity of a unit's weight polarity, we only ask the polarity configuration to stay fixed throughout training. Because we made fewer assumptions on the architecture and network properties, we were able to reveal the true power of weight polarities - polarity-fixed networks can not only perform as well as the traditional approaches when the polarities are set adequately, but they can also learn more quickly with less samples. Additionally, we revealed that polarities, not weights, may be a more effective and compressed form of knowledge transfer medium. Furthermore, our Freeze-SGD algorithm 1 is easily applicable to any existing network architecture and any learning mode, be it transfer learning or de novo learning, thus enjoy a wider application range.

Our hypothesis is also novel in the machine learning literature. In a statistical learning framework, it is counter-intuitive to disassemble a single parameter into magnitude and sign. To the best of our knowledge, we are the first to think in this way and prove that connection polarities themselves serve as an effective form of inductive bias. Previous work that vaguely took on a connectionist's view mostly focused on the existence or range of connections (e.g., adding skip connections (He et al., 2016)), but the polarity of such connections were essentially left out of the discussion. Our work broke the stereotypical view and brought weight polarity into the playing field. A lesson we learned from this research is that when designing network architectures, we should not only focus on the *existence* of connections, but also pay attention to the *polarities* of connections. In the transfer learning literature, to the best of our knowledge, there has been no indication that transferring polarity alone is sufficient for knowledge transfer – ours is the first demonstration on the importance of polarity.

Our work is in sync with the framework of learning theory. Our way of explicitly setting the weight polarities provides a strong prior for the networks. This is in line with the bias-variance trade-off theorem (Geman et al., 1992; Tibshirani & Friedman, 2001). Frozen-Net sufficient-Polarity has a high inductive bias – by trading representation capacity, they gain in time and data efficiency through discounting parameter variances. Indeed, the performance gain we observe from fixing polarities might be explained by preventing over-fitting thus better generalization to the validation data.

Our work is also in sync with the lottery ticket hypothesis (Frankle & Carbin, 2019). Our observations on larger networks enjoying a high probability of learning XOR-5D even when their polarities are randomly configured is a realization of the lottery ticket hypothesis.

We engineered our Freeze-SGD algorithm entirely based on SGD because we are interested in the pure effect of fixing polarity during learning. As discussed in Sec 2, such an approach intrinsically put Frozen-Net in a disadvantageous position as while resetting weights to the correct polarity after each learning iteration, this procedure effectively fights against the gradient update direction. It is an interesting next step to adopt a more polarity-freeze compatible learning algorithm, possibly allowing us to further improve learning performance. One possibility is adapting the primal-dual interior-point method (Vogelstein et al., 2010) or Hebbian learning (Amit, 2019), as well as a host of bio-plausible learning algorithms (Miconi, 2017; Boopathy & Fiete, 2022; Dellaferrera & Kreiman, 2022).

From this work, we have strong experimental evidence (simulation and image classification) and some theoretical discussion suggesting that if the weight polarities are adequately set *a priori*, then networks can learn with less data, time, space, and power. We look forward to applying our Frozen-Net approach to more tasks to see if we can empirically extend our results to more diverse scenarios. We also look forward to extending our results theoretically (e.g., including SGD assumption in our Lemma 2.1).

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

## A FREEZE-SGD ALGORITHM

---

**Algorithm 1 Freeze-SGD**

---

**for** $l = 1, 2, \ldots, L$ **do**
    Get weight polarity template $T^{(l)}$ based on the configuration rules. Match $W^{(l)}$ to $T^{(l)}$
**end for**

**for** $epoch = 1, 2, \ldots$ **do**
    **for** $batch = 1, 2, \ldots$ **do**
        SGD updates all weights.
        **for** $l = 1, 2, \ldots, L$ **do**
            Compare signs of the weights $W^{(l)}$ to the template $T^{(l)}$,
            get $checker = T^{(l)} * \text{sign}(W^{(l)})$.
            **for** $(i, j)$ where $T^{(l)}(i, j) < 0$ **do**
                Make $W_l(i, j)$ in compliance with $T^{(l)}(i, j)$ by one of the following four ways:
                **Case posCon** $W^{(l)}(i, j) = T^{(l)}(i, j) * \epsilon$ where $\epsilon > 0$
                **Case posRand** $W^{(l)}(i, j) = T^{(l)}(i, j) * rand([0, \epsilon])$ where $\epsilon > 0$
                **Case zero** $W^{(l)}(i, j) = 0$
                **Case flip** $W^{(l)}(i, j) = -W^{(l)}(i, j)$
            **end for**
        **end for**
    **end for**
**end for**

---

# B SUPPLEMENTARY FIGURES

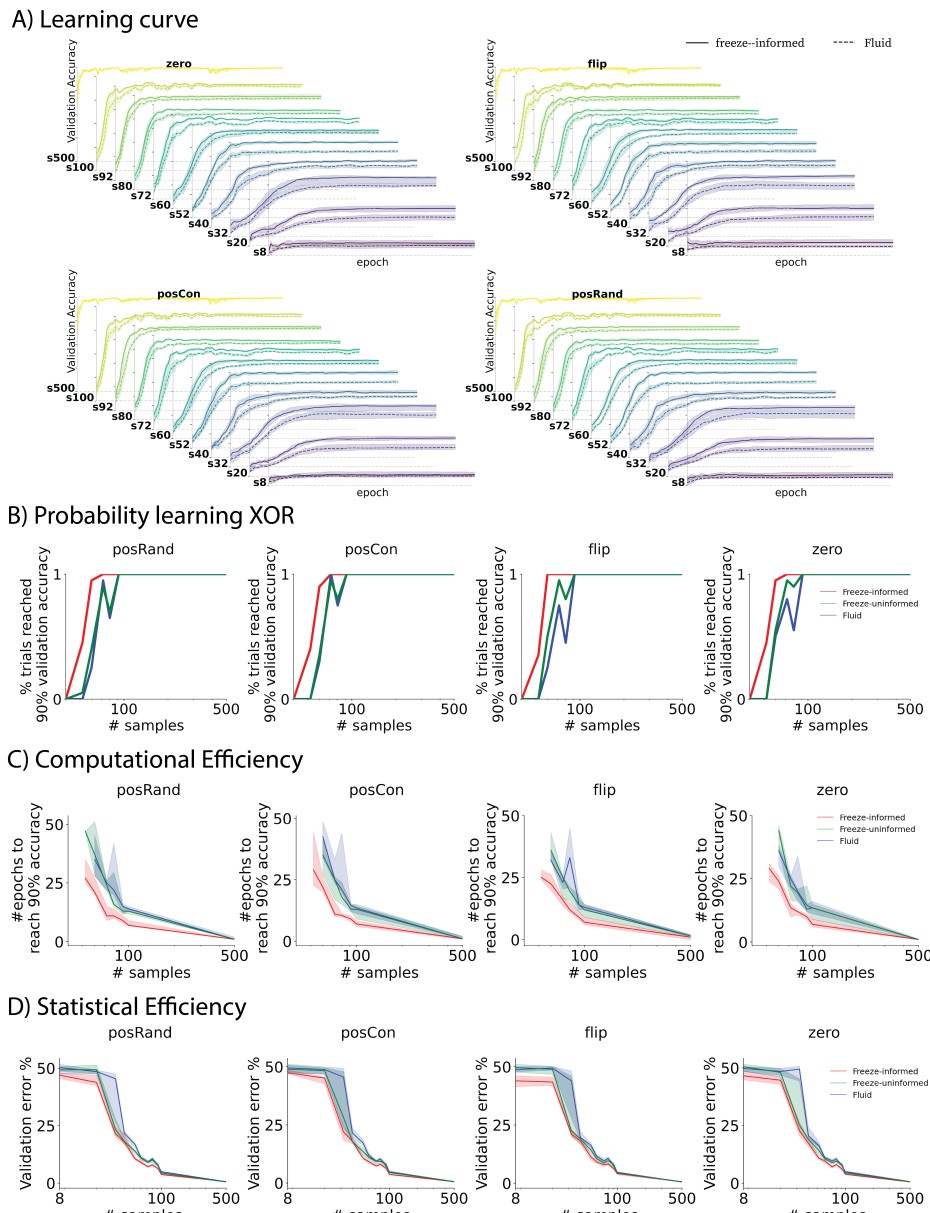

Figure B.1: **Different reset methods in the Freeze-SGD algorithm yields similar results.** The meaning of different reset methods is described in algorithm 1. A) Freeze-informed consistently reaches higher validation accuracy after the same amount of training time. This is especially evident when training sample is scarce. B) Same curves as in Fig 4, panels B & C first column, plotted for different reset methods. C) & D) Same curves as in Fig 1, panels B & C, plotted for different reset methods. We run 20 trials for all experiments in this figure. All curves correspond to medians with shaded regions representing the 25th and 75th percentiles.

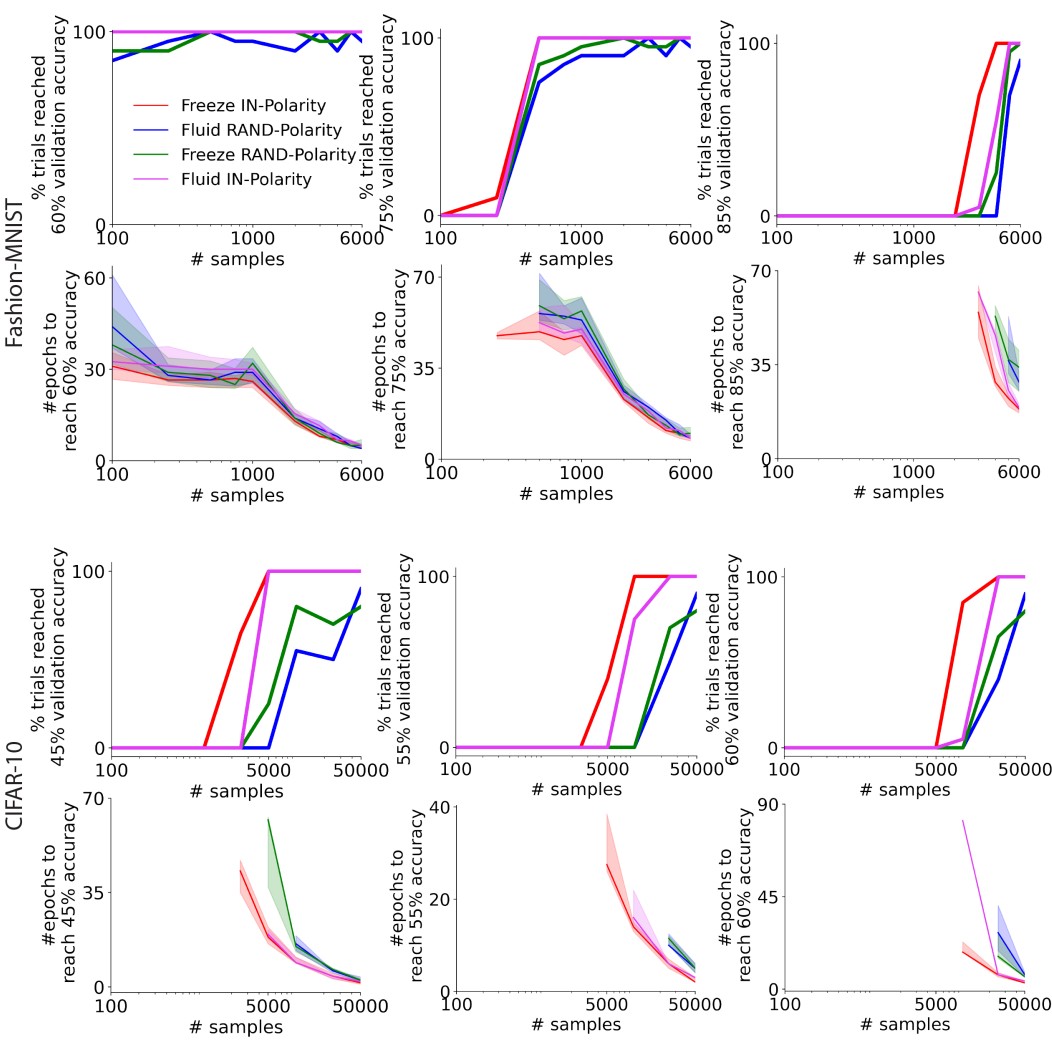

Figure B.2: **Related to Fig 2. Regardless of validation accuracy threshold, Frozen-Net IN-Polarity always learn more quickly.** Same curves as in Fig 2 right two columns, plotted for different validation accuracy thresholds.

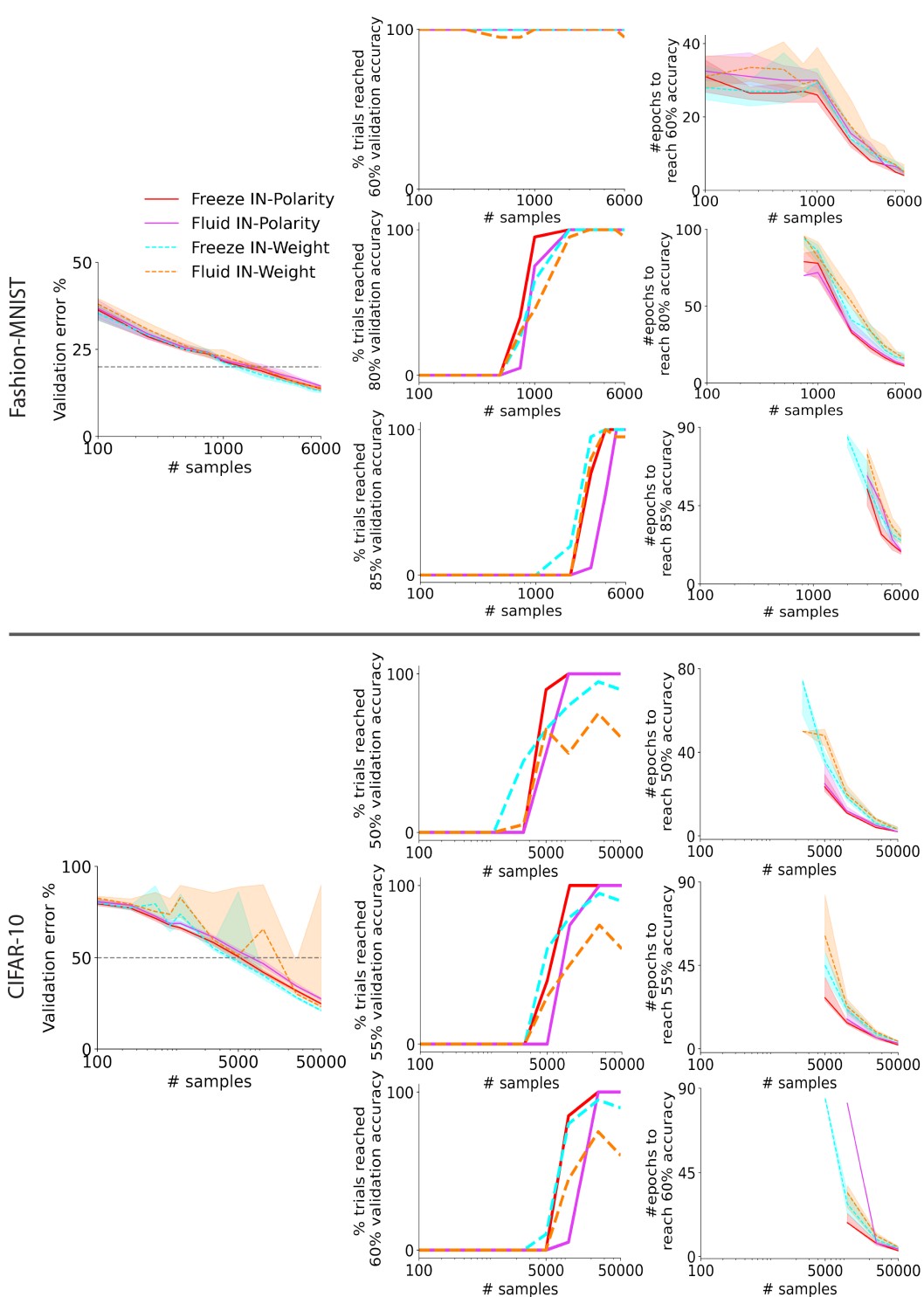

Figure B.3: **Related to Fig 3. All scenarios plotted separately instead of plotting only for the differences.**

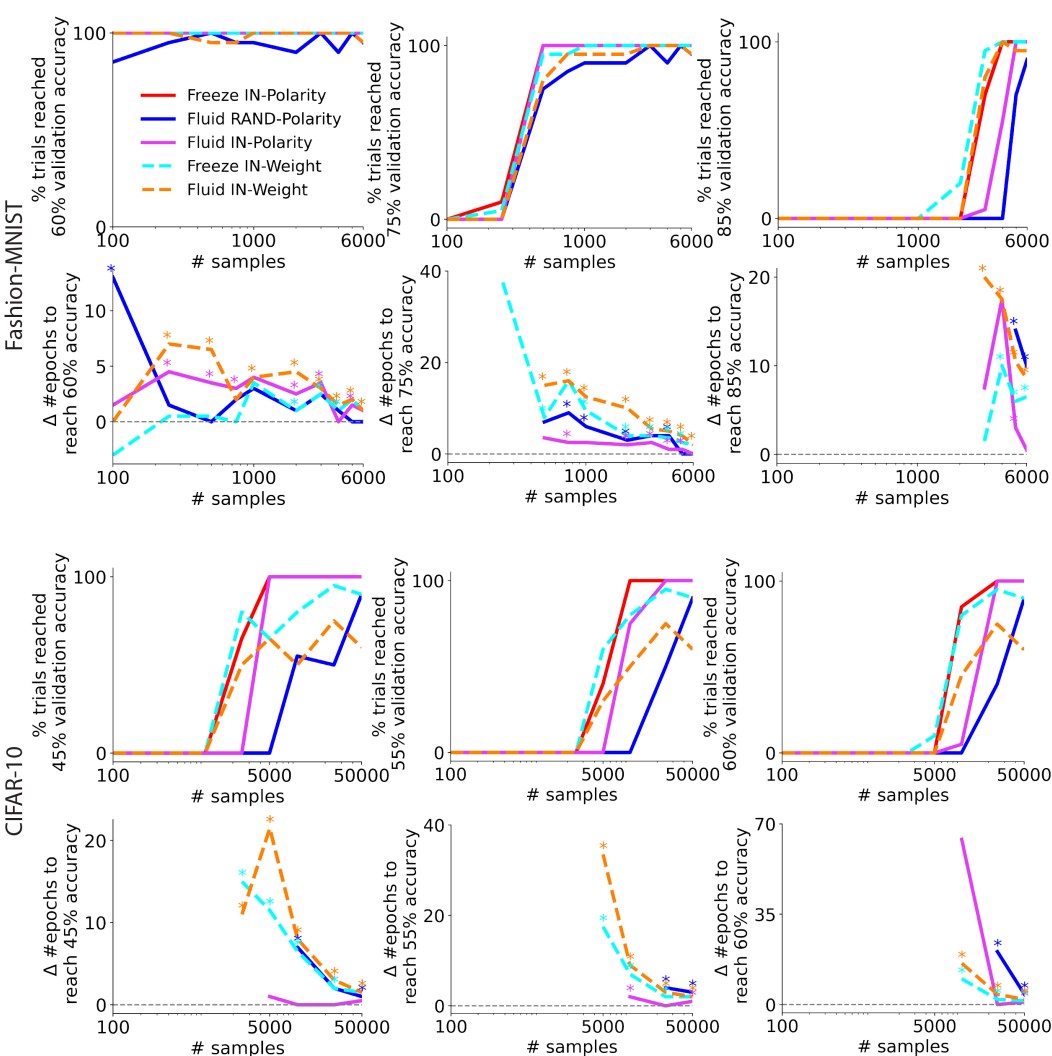

Figure B.4: **Related to Fig 3. Regardless of validation accuracy threshold, transferring and fixing polarities help networks learn faster than traditional weight transfer strategy.** Same curves as in Fig 3 right two columns, plotted for different validation accuracy thresholds.

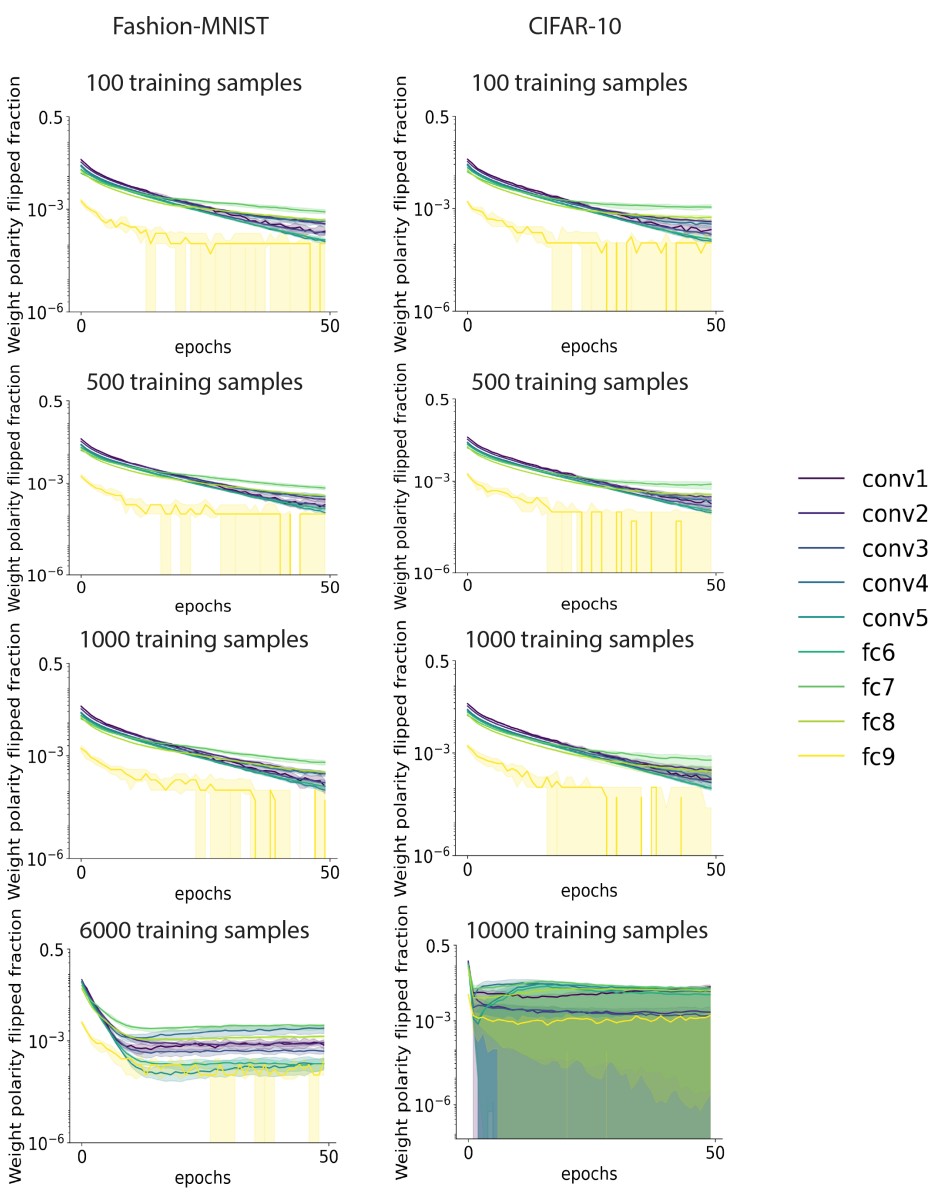

Figure B.5: **Tracking number of polarity flips.** Weight polarity flips were analyzed for Fluid RAND-Polarity (SGD with random initialization) and measured by the ratio of weight parameters (excluding bias terms) flipped sign between two consecutive epochs. The first 50 epochs were analyzed and plotted, separately across layers and training data size. Curves are median with shaded area representing the 25th and 75th percentiles out of the 20 trials.

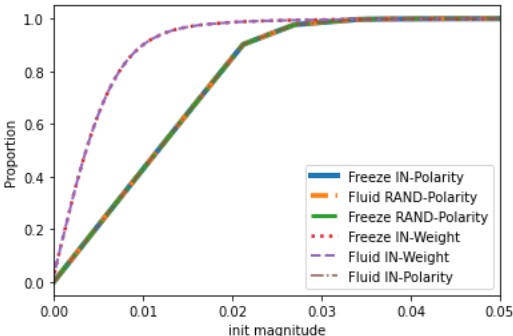

Figure B.6: **Weight magnitude distribution at initialization across all experimental conditions for image classification tasks.** All scenarios follow the exact same magnitude initialization magnitude except IN-weight transfer.

## C  METHODS

**XOR-5D**  Data were prepared by sampling from the XOR-5D distribution as described in the main text, the training data size varied, validation set is always 1000 samples across all scenarios. A single hidden layer network (64 hidden units for Fig 1, various for Fig 4) was trained for 100 epochs; in total 50 randomly seeded initializations (trials) were ran. To have the best controlled experiments, we fixed the magnitude distribution, architecture, training samples (n varies), learning rate, batch sequence of the training data, and the validation samples across all tested scenarios. **Statistical efficiency** was quantified by plotting validation error rate *at convergence* across different training data scarcity levels. **Computational efficiency** was quantified by plotting the number of epochs to reach certain level of validation accuracy. We note not all trails could reach the same level of validation accuracy cut-off, to present the whole picture, we 1) plotted the percentage of trials that reached the validation accuracy cut-off (**success rate**); 2) showed results across a range of cut-off selections (supp Fig B.2, B.3, B.4).

**Image classification**  Experiments were done essentially the same as XOR-5D in a tightly controlled fashion, same magnitude distribution, batch sequence (batch size = 1000), learning rate, and training data across all scenarios. lr=0.001 was chosen from [0.03, 0.01, 0.005] based on the most accuracy gain after 50 epochs across all scenarios. Networks were trained for 100 epochs, 20 trials in total. We did not do any image augmentation. All performance quantification follow the above description.

All experiments were run on 2 RTX-8000 GPUs.

# D  XOR THEOREMS RELATED TO SEC 5

**Lemma D.1** (minimum XOR solution, 2D). *For any single-hidden-layer network to be able to solve XOR, it is sufficient to have 3 different XOR-compatible hidden units. The weight pattern for each unit can be given by a triplet $(w_{1,j}^{(1)}, w_{2,j}^{(1)}, w_{j,1}^{(2)})$, where $j$ is the unit index, $j \in \{1, \ldots, n\}$, $n$ is the number of hidden units. An XOR compatible hidden unit is one where the weight pattern triplet have their polarities satisfy $sign(w_{j,1}^{(2)}) = sign(w_{1,j}^{(1)}) \times sign(w_{2,j}^{(1)})$.*

See proof on page 20.

**Lemma D.2** (can learn XOR probability, 2D). *For a single hidden layer network with $n$ hidden units $(n \in \mathbb{Z}^+)$, randomly initialize each weight with its polarity following $P(polarity) \sim Bernoulli(0.5)$, then the probability that this randomly initialized network can learn XOR without changing any of the weight polarity is lower bounded by*

$$\Omega(P(n)) = 1 - \frac{\sum_{k=0}^{2} Q(k, 4, 4, n)}{8^n}, \quad n \in \mathbb{Z}^+$$

$$Q(k, m, M, n) = [(M + k)^n - \sum_{p=0}^{k-1} Q(p, k, M, n)]\binom{m}{k}$$

*Q here is a helper function for counting, it is defined in more detail in the proof.*

See proof on page 21.

**Theorem D.1** (can learn XOR probability, high dimensional). *For a single hidden layer network with $n$ hidden units, randomly initialize each weight with its polarity following $P(polarity) \sim Bernoulli(0.5)$, then the probability that this randomly initialized network can learn high-dimensional XOR (first two dimensions are relevant, the rest $(d - 2)$ dimensions are irrelevant) without changing any of the weight polarities is lower bounded by*

$$\Omega(P(n, d)) = 1 - \frac{\sum_{k=0}^{2} Q(k, 2^d, 2^d, n)}{(2^{d+1})^n}, \quad n, d \in \mathbb{Z}^+, d \geq 2$$

$$Q(k, m, M, n) = [(M + k)^n - \sum_{p=0}^{k-1} Q(p, k, M, n)]\binom{m}{k}$$

*Q here is a helper function for counting, it is defined in more detail in the proof.*

See proof on page 22.

# E  PROOFS

**Lemma 2.1** (capacity-speed trade-off). *If the weight polarities are set a priori, such that the function is still representable, then the network can learn faster.*

*Proof of Lemma 2.1.* A feedforward DNN can be described by as a graph

$$G = (V, E), w : E \to \mathbb{D}.$$

Nodes of the graph correspond to neurons (units), where each neuron is a function $\sigma_j^{(l)}(x) = \sigma(W_j^{(l)}x + b^{(l)}), j \in [n_l]$. All the weights for the network take on values from the set $\mathbb{D} = \{-d, \ldots, 0, \ldots, d\}$ for some $d \in \mathbb{N}$. The network is organized in *layers*. That is, the set of nodes can be decomposed into a union of (nonempty) disjoint subsets, $V = \dot{\bigcup}_{l=0}^{L} V_l$, such that every edge in $E$ connects some node in $V_{l-1}$ to some node in $V_l$, for some $l \in [L]$. Assume we have fully connected layers. Then the number of incoming edges per node is $|V_{l-1}|$. Let $|V_0|$ be the input space dimensionality. All nodes in $V_0$ (input layer) and $V_L$ (output layer) are distinct.

Let $|G|$ denote the total number of distinct weight patterns of the graph $G$. Then for a single hidden layer network where $L = 2$, we have:

$$|G| = |\mathbb{D}|^{|E|} = |\mathbb{D}|^{(|V_0|*|V_1|+|V_1|*|V_2|)} \tag{2}$$

Then

$$\frac{|G|}{|G_{polarityFrozen}|} = \left(\frac{2d+1}{d+1}\right)^{(|V_0|*|V_1|+|V_1|*|V_2|)} \tag{3}$$

Assume a different weight pattern represents a different function (trivially holds in linear + full rank weight case), then for every single representable function, there always exists a set of weight polarity configurations $G_{correct}$ such that $G_{correct} \subseteq G_{polarityFrozen}$, therefore

$$\frac{|G_{correct}|}{|G_{\text{polarityFrozen}}|} \ll \frac{|G_{correct}|}{|G|} \tag{4}$$

This means setting weight polarity *a priori* in an adequate way constraints the combinatorial search space to have much higher proportion of correct solutions, hence easier to learn under exhaustive search algorithm (Lemma 2.1). □

**Lemma D.1** (minimum XOR solution, 2D). *For any single-hidden-layer network to be able to solve XOR, it is sufficient to have 3 different XOR-compatible hidden units. The weight pattern for each unit can be given by a triplet $(w_{1,j}^{(1)}, w_{2,j}^{(1)}, w_{j,1}^{(2)})$, where $j$ is the unit index, $j \in \{1, \ldots, n\}$, $n$ is the number of hidden units. An XOR compatible hidden unit is one where the weight pattern triplet have their polarities satisfy $sign(w_{j,1}^{(2)}) = sign(w_{1,j}^{(1)}) \times sign(w_{2,j}^{(1)})$.*

*Proof of Lemma D.1.* For each hidden unit, we can enumerate all of its 8 possible weight polarity configurations, with the index set $A_{Polarities} = \{1, 2, 3, 4, 5, 6, 7, 8\}$:

| # | $w_{1,j}^{(1)}$ | $w_{2,j}^{(1)}$ | $w_{j,1}^{(2)}$ |
|---|---|---|---|
| 1 | + | + | + |
| 2 | + | + | - |
| 3 | + | - | + |
| 4 | + | - | - |
| 5 | - | + | + |
| 6 | - | + | - |
| 7 | - | - | + |
| 8 | - | - | - |

The set $B_{XOR-Polarities} = \{2, 3, 5, 8\} \subset A_{Polarities}$ contains indexes of polarity patterns that follow XOR. To prove Lemma D.1, we first consider the case of 3 hidden unit single layer network. To prove a network of certain polarity pattern is capable of solving XOR, it suffices to give a working solution. Below, we exhaust all possible 3-unit network that satisfy the statement of Lemma D.1, i.e. having 3 different XOR-compatible units; and list out their corresponding working network solutions.

| {} | # | $w_{1,j}^{(1)}$ | $w_{2,j}^{(1)}$ | $b_j^{(1)}$ | $w_{j,1}^{(2)}$ | working network $F(x,y)$= |
|---|---|---|---|---|---|---|
| | 2 | +1 | +1 | 0 | -1 | |
| {2,3,5} | 3 | 1 | 0 | 0 | 1 | $\text{sigmoid}(-\sigma(x+y)+\sigma(x)+\sigma(y))$ |
| | 5 | 0 | 1 | 0 | 1 | |
| | 2 | +1 | 0 | 0 | -1 | |
| {2,3,8} | 3 | +1 | -1 | 0 | +1 | $\text{sigmoid}(-\sigma(x)+\sigma(x-y)-\sigma(-y))$ |
| | 8 | 0 | -1 | 0 | -1 | |
| | 2 | 0 | +1 | 0 | -1 | |
| {2,5,8} | 5 | -1 | +1 | 0 | +1 | $\text{sigmoid}(-\sigma(y)+\sigma(y-x)-\sigma(-x))$ |
| | 8 | -1 | 0 | 0 | -1 | |
| | 3 | 0 | -1 | 0 | +1 | |
| {3,5,8} | 5 | -1 | 0 | 0 | +1 | $\text{sigmoid}(\sigma(-y)+\sigma(-x)-\sigma(-x-y))$ |
| | 8 | -1 | -1 | 0 | -1 | |

For the case that single-hidden-layer network has more than 3 hidden units, if it satisfies the rule in Lemma D.1, we can always construct a XOR-solvable network solution by setting all other weights to zero except for 3 different XOR-compatible units, then we arrive at one of the four situations in the above table and we just proved they are XOR solutions.

Therefore, for any single hidden layer network to solve XOR, it is sufficient to have at least 3 of the 4 XOR polarity patterned units. □

**Lemma D.2** (can learn XOR probability, 2D). *For a single hidden layer network with $n$ hidden units $(n \in \mathbb{Z}^+)$, randomly initialize each weight with its polarity following $P(polarity) \sim Bernoulli(0.5)$, then the probability that this randomly initialized network can learn XOR without changing any of the weight polarity is lower bounded by*

$$\Omega(P(n)) = 1 - \frac{\sum_{k=0}^{2} Q(k,4,4,n)}{8^n}, \quad n \in \mathbb{Z}^+$$

$$Q(k,m,M,n) = [(M+k)^n - \sum_{p=0}^{k-1} Q(p,k,M,n)]\binom{m}{k}$$

*Q here is a helper function for counting, it is defined in more detail in the proof.*

*Proof of Lemma D.2.* For any network randomly initialized, its weight polarity pattern is a set of $n$ draws with replacement from the polarities indexed by the set $A_{Polarities}$, $|A_{Polarities}| = 8$. Define $H = (h_1, \ldots, h_m, \ldots, h_n), h_m \in A_{Polarities}, |H| = n$, to be the tuple of the indices of the observed weight patterns for the hidden layer. $h_m$ is the index of the weight polarity pattern for unit $m$. For any network to be able to solve XOR, it needs to have at least 3 units whose weight patterns are distinct members of set $B_{XOR-Polarities}$ (Lemma D.1). That is, let $J = \{h_m : h_m \in H, h_m \in B_{XOR-Polarities}\}$. Then we need that $|J| \geq 3$. We can define the probability of having exactly $k$ of the 4 XOR compatible weight patterns present within the hidden layer as following (for brevity, $A = A_{Polarities}, B = B_{XOR-Polarities}$):
None of the members in set B appears is given by:

$$P(|J| = 0) = \frac{|B|^n}{|A|^n} = (\frac{4}{8})^n = (\frac{1}{2})^n$$

Only one of the member in set B appeared in H, and that member can appear more than once in H is given by:

$$P(|J| = 1) = \frac{\binom{|B|}{1}((|B|+1)^n - |B|^n)}{|A|^n} = \frac{\binom{4}{1}(5^n - 4^n)}{8^n}$$

This is explained by choosing one of 4 members of B $\binom{|B|}{1}$, then multiply by the chosen member appears at least once $((|B|+1)^n - |B|^n)$

Only two of the members in set B appeared in H, and both can appear more than once in H:

$$P(|J| = 2) = \frac{\binom{|B|}{2}((|B| + 2)^n - \binom{2}{1}((|B| + 1)^n - |B|^n) - |B|^n)}{|A|^n}$$

$$= \frac{\binom{4}{2}(6^n - \binom{2}{1}(5^n - 4^n) - 4^n)}{8^n}$$

To count two members appear at least once in $H$, we have $\binom{|B|}{2}$ ways of choosing the 2 members from set $B$, and there are $(|B| + 2)^n$ ways of choosing with replacement to populate the tuple $H$, where each hidden unit can choose from in total $(|B| + 2)$ possible patterns, and subtract the situation where only one of the two members appeared $\binom{2}{1}((|B| + 1)^n - |B|^n)$ and the situation where neither appeared $|B|^n$

The above equations can be put into a compact form

$$P(|J| = k) = \frac{Q(k, |B|, |A| - |B|, n)}{|A|^n}$$

where $Q(k, m, M, n) = [(M + k)^n - \sum_{p=0}^{k-1} Q(p, k, M, n)]\binom{m}{k}, k \in \{0, \ldots, m\}, m \in 1, \ldots, |B|, M \in 1, \ldots, |A| - |B|, n \in \mathbb{Z}^+$ is a helper function that gives exactly $k$ of the $m$ different XOR-compatible polarity patterns appeared in $n$ units, and in total $m + M$ options are considered for each unit.

Then, $P(n)$ is counting at least 3 of the 4 set $B$ patterns appear:

$$\Omega(P(n)) = P(|J| \geq 3) = 1 - P(|J| = 0) - P(|J| = 1) - P(|J| = 2)$$

$$= 1 - \frac{\sum_{k=0}^{2} Q(k, 4, 4, n)}{8^n}$$

$$= 1 - (\frac{1}{2})^n - \frac{\binom{4}{1}(5^n - 4^n)}{8^n} - \frac{\binom{4}{2}(6^n - \binom{2}{1}(5^n - 4^n) - 4^n)}{8^n}.$$

$\square$

**Theorem D.1** (can learn XOR probability, high dimensional). *For a single hidden layer network with $n$ hidden units, randomly initialize each weight with its polarity following $P(polarity) \sim Bernoulli(0.5)$, then the probability that this randomly initialized network can learn high-dimensional XOR (first two dimensions are relevant, the rest $(d - 2)$ dimensions are irrelevant) without changing any of the weight polarities is lower bounded by*

$$\Omega(P(n, d)) = 1 - \frac{\sum_{k=0}^{2} Q(k, 2^d, 2^d, n)}{(2^{d+1})^n}, \quad n, d \in \mathbb{Z}^+, d \geq 2$$

$$Q(k, m, M, n) = [(M + k)^n - \sum_{p=0}^{k-1} Q(p, k, M, n)]\binom{m}{k}$$

*Q here is a helper function for counting, it is defined in more detail in the proof.*

*Proof of Theorem D.1.* We solve Theorem D.1 with the exact same counting algorithm as in Lemma D.2, the only difference is now $|A_{Polarities}| = 2^{d+1}$ and $|B_{XOR-Polarities}| = 2^d$. We prove these two equalities below.

$|A_{Polarities}| = 2^{d+1}$ because we have $d$ input weights and 1 output weight for each unit.

The weight pattern of a single hidden unit in this case is given by a tuple $(w_{1,j}^{(1)}, \ldots, w_{d,j}^{(1)}, w_{j,1}^{(2)}), j \in \{1, \ldots, n\}$ and $n$ is the number of hidden units. Our conclusion in Lemma D.1 trivially extends to the high dimensional case. This is because high-dimensional-XOR-solvable network solutions can be trivially constructed from Lemma D.1 by setting the irrelevant input dimension weights to 0. We can restate Lemma D.1 for high-dimensional XOR as following:

For any single-hidden-layer network to be able to solve high dimensional XOR (only first two dimensions are relevant), it is sufficient to have 3 different XOR-compatible hidden units. A high dimensional XOR compatible unit polarity pattern is ruled by $sign(w_{j,1}^{(2)}) = sign(w_{1,j}^{(1)}) \times sign(w_{2,j}^{(1)})$.

Therefore the irrelevant dimension input weights can be of either polarities and there are $2^{d-2}$ different combinations of them. Therefore $|B_{XOR-Polarities}| = 4 \times 2^{d-2} = 2^d$.

Follow the steps in Lemma D.2, we have (for brevity, $A = A_{Polarities}, B = B_{XOR-Polarities}$)

$$P(|J| = 0) = \frac{|B|^n}{|A|^n} = (\frac{2^d}{2^{d+1}})^n$$

$$P(|J| = 1) = \frac{\binom{|B|}{1}((|B|+1)^n - |B|^n)}{|A|^n} = \frac{\binom{2^d}{1}((2^d+1)^n - (2^d)^n)}{(2^{d+1})^n}$$

$$P(|J| = 2) = \frac{\binom{|B|}{2}((|B|+2)^n - \binom{2}{1}((|B|+1)^n - |B|^n) - |B|^n)}{|A|^n}$$

$$= \frac{\binom{2^d}{2}((2^d+2)^n - \binom{2}{1}((2^d+1)^n - (2^d)^n) - (2^d)^n)}{(2^{d+1})^n}$$

Therefore, we have

$$\Omega(P(n,d)) = 1 - \frac{\sum_{k=0}^{2} Q(k, 2^d, 2^d, n)}{(2^{d+1})^n}$$

$$= 1 - (\frac{1}{2})^n - \frac{\binom{2^d}{1}((2^d+1)^n - (2^d)^n)}{(2^{d+1})^n} - \frac{\binom{2^d}{2}((2^d+2)^n - \binom{2}{1}((2^d+1)^n - (2^d)^n) - (2^d)^n)}{(2^{d+1})^n},$$

$$n, d \in \mathbb{Z}^+, d \geq 2$$

$\square$

