# OpenReview forum: "Polarity is all you need to learn and transfer faster"
_ICLR.cc/2023/Conference — Submitted to ICLR 2023_

### Official Review · Reviewer_8Q8m · 2022-10-23

**Confidence:** 4
**Correctness:** 2
**Technical Novelty And Significance:** 2
**Empirical Novelty And Significance:** 3
**Recommendation:** 3

**Clarity, Quality, Novelty And Reproducibility:**

Clarity: The writing in the paper is generally clear, except for the vagueness around many of the claims w.r.t. natural intelligence and biological neural networks.

Quality:
 * Neuroscientific motivation, connections and learnings - poor.
 * Experiments: some important ablations missing to solidify the main claims and rule out alternative hypotheses, but multiple repetitions of runs and stat. significance checks - good.
 * Theory - ok but not very relevant (because of limitation to exhaustive search) - poor.
 * Discussion of related work: very long and raises many connections, overall rather superficial though and could be strengthened by being more precise - ok.

Novelty: the main technical idea and experiments in the paper are novel to the best of my knowledge.

Reproducibility: Have not checked carefully but some minor experimental details might be missing; maybe worth doing another pass on this, but nothing of major concern.


**Strength And Weaknesses:**

**Main contributions, Impact**
1) Empirical analysis of the gains of freezing weight polarity during training using oracle knowledge. To the best of my knowledge this is novel. Results confirm the expectation that additional knowledge helps reduce the required number of training examples / increases sample efficiency by providing additional biases. While the result is interesting, its practicality is very limited since this oracle knowledge is often unavailable.

2) Empirical analysis of using pre-trained signs instead of pre-trained weights for adapting from ImageNet to FashionMNIST/CIFAR-10. In the very low data-regime, where the model still has very large validation errors, there is some small, but consistent improvement (Fig. A.3 gives a better sense of the scale). This could potentially be interesting, but is currently of very limited impact due to the following two issues: (i) effect currently only holds in a regime where the model performs very poorly and disappears for large enough sample sizes from the transfer dataset; (ii) current experiments cannot rule out that we simply observe a regularizer that prevents overfitting to the small dataset instead of transferring useful knowledge (the missing control experiment for this is Freeze RAND-polarity).

3) Simple theoretical analysis that fixing the polarity of weights reduces the cardinality of the space of possible representable functions. This trivially leads to a reduced search time under exhaustive search. Impact: marginal - the result is very simple to construct, and its relevance for training a network via SGD is unclear (SGD is different from exhaustive search and a reduction in cardinality of the hypothesis space does not trivially translate into faster and more sample efficient SGD training).

**Strengths**
 * Investigation of an effect not studied before (to the best of my knowledge)
 * Most of the paper is written clearly
 * Statistical significance analysis of some results

**Weaknesses**
 * Practical relevance currently seems minor: in many cases oracle knowledge around the correct polarity is not available, and the current transfer learning results are not very convincing (only small improvements in a regime where the performance of the classifier is quite bad).
 * Theoretical analysis has no link to training neural networks via SGD.
 * Connections and claims w.r.t. natural intelligence, learning in biological brains, and neuroscience are overstated, vague, and of insufficient scholarly standard. What’s needed as a minimum is a technical discussion of learning in dynamic models of neural processing (spiking neurons or other models involving short- and long-term neuronal **dynamics**); talking about the role of excitatory and inhibitory synapses by referring to the sign of weights in an extremely simplified rate-based model only allows for extremely simple statements - the paper makes claims far beyond these. This would be tolerable if toned down and limited to a small paragraph in the discussion, but currently occupies roughly a third of the paper throughout all sections.

**Improvements**

1) Remove the claims and analogies w.r.t. biological brains and how this paper uncovers the main reason why evolutionary optimization has led to fixed polarity of “neuronal connections”. It is ok to use this fact about biological neural networks as an inspiration (one, two sentences in the intro) and then have one paragraph of discussion, but none of the current discussion is actually necessary for the technical part concerning artificial neural networks and SGD. If the amount of space devoted to this is kept, then the scholarly standard needs to be raised significantly. As a starting point I strongly suggest reading up on *spiking neuron models* and various types of *short-term plasticity*, as well as neuro-computational models using circuits of spiking neurons and population dynamics. Many basic textbooks in computational neuroscience cover the topic - I can recommend “Spiking Neuron Models: Single Neurons, Populations, Plasticity” by Gerstner and Kistler 2002. To defend against this criticism raised during the last round of reviews the paper says “As our goal is to see the pure effect of fixing weight polarity, we did not adopt any bio-plausible learning algorithms as they may introduce confounding factors.” - this is fine and reasonable, as long as **NO** claims are made w.r.t. biological learning or brains. If such claims are made they need to be discussed in the context of biologically plausible neuron models and learning algorithms.



2) Important questions w.r.t. the neuroscientific connections currently not answered in the paper:
  * Biological neural networks are much more dynamic than artificial neural networks (= simple rate based models). Neurotransmitters, short-term plasticity, electrochemical depletion, all the way to the dynamic recruitment of whole neuronal circuits and populations can play a huge role in the function of the brain for rapid adaptation (few show learning) and transfer learning - and different excitatory and inhibitory mechanisms play a significant role, and have been well studied, in each of these. The paper currently only says “neuronal connections in the brain” - what exactly does that refer to (synapses, neurons, populations?), and what exact excitatory and inhibitory mechanisms is the paper trying to explain?
  * The experiments in the paper use artificial neural networks (= simple rate based models) trained via SGD. Most neuroscientists would claim that results obtained from these have very limited bearing on explaining the function of biological neural networks and biological learning. Why do the experiments in this paper have biological relevance, and what exactly is that relevance?
  * The one neuroscientific reference provided in the paper (Spitzer 2017), except for the discussion, seems quite irrelevant - it is about the (rare?) phenomenon of ‘neurotransmitter switching’ where “Neurotransmitter receptors on postsynaptic cells change to match the identity of the newly expressed neurotransmitter.” [from the paper’s abstract]. While the phenomenon is potentially associated with certain neural diseases and addiction, it is not hypothesized to have a connection with few-shot learning or transfer learning. Why choose this as the one reference?
  * “post-development, neuronal connections in the brain rarely see polarity switch”. What developmental stage is exactly meant here? What “connections” are meant? Be precise, and add citations.
* “What then makes it so that our brains may have willingly chosen to give up on a vast portion of their representation capacity? Our answer is: to learn more quickly.” I disagree with this answer. The widely agreed upon reason (to the best of my knowledge) is that biological (spiking!) neurons communicate activations via increases in firing rate; without inhibitory connections stable dynamics in circuits of such neurons are impossible (activations, and thus firing rates would reinforce each other without bounds, which quickly drives neurons beyond the biologically feasible operating regime; there are hard physical limitations for maximum firing rates of neurons and neural populations - inhibitory mechanisms are necessary to allow any computation at all). Please clarify.

3) Figure 2 and 3 also need to show Liquid (IN-)polarity, where the correct polarity (but not the weight magnitudes for Fig 3) are used to initialize the network, but then the polarity is free to change during training. This answers whether it is important to freeze the polarity (as proposed in the paper), or whether it’s sufficient to only initialize it correctly.

4) Fig. 3 is missing the Freeze RAND-polarity experiment, where the correct proportion of polarities is frozen but not necessarily for the correct connections. This allows ruling out that  freezing polarity simply provides additional regularization but does not lead to any transfer of information from the pre-trained network.

5) Perform the transfer learning analysis in Fig. 3 layer-wise (i.e. transfer polarity for the first layer only, then the first two layers, etc). One hypothesis to explain the results in the paper is that the transfer of Gabor-like filters in very early layers explains a large fraction of the observed performance gains. Sign-information might indeed be virtually all that’s needed for Gabor-like filters. Transferring whole weights for the final classification layer might cause most of the observed decrease in performance for Liquid IN-weight.

6) Ideally repeat Fig. 3 with a validation set and early stopping instead of a fixed set of epochs, to rule out effects of overfitting on the very small datasets.


**Minor comments**

A) Change in notation from ‘Fluid’ (Fig 2) to ‘Liquid’ (Fig 3).

B) “In the literature of bio-plausible artificial neural networks (ANNs), the most related work is on Dale’s principle: a single unit’s output weights are exclusively excitatory or inhibitory (Dale, 1935).” With all due respect, but the literature on biologically plausible neural networks was at best at its infancy in 1935 - if this is the closest reference to this work, I would see this as a strong sign of concern in terms of biological plausibility (not a sign of groundbreaking novelty of the current work).

C) “Our angle is completely novel in the neuroscience literature.” This is too strong; there is a vast body of literature on the relationship between excitatory and exhibitory circuits/neurons/synapses/neurotransmitters and their role e.g. in decision-making and short-term adaptation and motor control. While the core idea of fixed polarity might be an interesting hypothesis to explore seriously from a modern neural dynamics perspective, the current work fails to bridge this gap.

D) “As discussed in Sec 2, such an approach intrinsically put Frozen-Net in a disadvantageous position”. This is not entirely correct: the net is in a disadvantageous position in terms of representing any possible function - if the polarity is set correctly the net is in a highly advantageous position since the search-problem has been greatly reduced (under exhaustive search).


**Summary Of The Paper:**

**Update after rebuttal**
The authors added an important control experiment (fluid in-polarity) which shows that the main improvements result from transferring a favorable initial set of weights, whereas actually freezing the polarity (which is *the* main NI inspiration) has a a marginal effect (and it is conceivable that this effect mainly results from very early layers). This makes the empirical results in the paper stronger, but significantly weakens the link to the NI inspiration (after all, as the authors point out, polarity change in biological brains is rare). The authors chose to largely keep the emphasis on NI and reformulate one, two sentences (where explicit claims were made) rather than shortening the discussion to a single paragraph in the discussion - this is a rhetoric trick to save work with re-writing, but essentially leaves most of my criticism unaddressed. Overall, the authors rebuttal was quite deflective and many of the issues I raised were not addressed or only minimally addressed (see detailed comments to author feedback). I therefore remain with my final verdict, that at its core the paper explores an interesting phenomenon in transfer learning (a combination of good initialization and a regularizing effect of fixing polarity); the connections to NI are misleading and contradictory (if freezing polarity is so important, how can the fluid-in-polarity results be explained; in fact it is currently unclear what fraction of weights changes polarity in that setting). From a purely transfer learning perspective it is unclear whether the observed effects also hold in a regime where models are practically useful (currently results where the proposed scheme performs well are in the low accuracy regime); limiting the impact of the work to practitioners aiming to improve classifiers' performance beyond the very low data regime. Given the additional control experiments and some of the small fixes I would increase my score by one point (from 3 to 4), but the current scoring system only allows a 5 as the next score and I personally think that the paper in its current form is not borderline. I remain open to updating my final verdict based on the reviewer discussion of course.


The paper investigates the effect of fixing the signs of weights (polarity) in an artificial neural network during training by using an oracle (expert knowledge or pre-trained network). This idea is inspired by the fact that “neuronal connections in the brain rarely see polarity switch”. As the paper shows on some simple datasets, if signs of weights are set correctly (and not randomly) and held fixed during training, faster / more sample-efficient learning is possible compared to a network that does not use the oracle knowledge (i.e. a standard neural network). Some theoretical insight is provided via the argument that fixed polarity leads to a reduction over the functions representable by the network, and thus a reduced search space over functions via exhaustive search. The second set of experiments shows that transfer learning in the very low data regime with pre-trained ImageNet weights to FashionMNIST and CIFAR-10 works slightly (but statistically significantly) better when only transferring weight polarity but not magnitude.

Disclaimer: I have reviewed a previous version of this manuscript, and am happy to see that the authors have taken into account some suggestions raised by the reviewers, but have chosen to ignore some other suggestions. My review might thus be somewhat repetitive in some places (I have not recycled anything from my previous review though).


**Summary Of The Review:**

Overall I think the paper has an interesting technical core (investigating the effect of frozen weight polarity, provided by an oracle / pre-trained net). There are some alternative hypotheses that need to be ruled out via control experiments (see improvements) and the results are currently a bit limited in terms of practical relevance. Particularly, the transfer learning results would be very interesting if observed in more challenging settings and in a regime where the classifier actually performs reasonably well. The theory does not add much to the paper unless it is extended to include SGD, which currently does not seem trivial. In my opinion, the neuroscientific/neuroevolutionary speculation currently lowers the overall quality of the paper - I suggest removing it completely from the current work, since it is not necessary and does not add to the technical part on artificial neural networks trained via SGD. As the paper currently stands I recommend rejection - even if the neuroscientific motivation is removed, the results (though promising) are not quite above the threshold for an ICLR publication (but would make an interesting workshop contribution). Having said that, I think compared to the version I previously reviewed, the paper has made good progress in the right direction.

---

> ### Author Response · Authors · 2022-11-19
> **Constructive feedback all long and thank you for recognizing the improvements we've made!**
>
> We thank reviewer $\Qm$ for the constructive feedback all long and recognizing the improvements we've made!
>
> **While the result is interesting, its practicality is very limited since this oracle knowledge is often unavailable.**
>
> Oracle knowledge exists to its largest extent in transfer learning. Our observation is simply that transfer learning appears in some cases to be more efficient when the network retraining is constrained to keep the same polarity structure. Imposing this constraint does indeed require access to a network previously trained on a sufficiently similar task to have learned a polarity structure sufficient to learn the new task.
>
> **Effect currently only holds in a regime where the model performs very poorly and disappears for large enough sample sizes from the transfer dataset**
>
> We agree with the observation but disagree with the implied significance of our work. We are particularly interested in the small sample size cases because we this is the gap between natural and artificial intelligences we aim to close. To the best of our knowledge, we were not aware of an existing bio-plausible AI feed-forward DNN work that demonstrated more than matching the current state performance - we improved on the small sample regime where AI struggle but NI excel. We agree it is an interesting future direction to further close this gap.
>
> **Missing control experiment Freeze RAND-polarity**
>
> We would like to point out that Freeze RAND-Polarity $\textbf{is included}$ in both our experiment on XOR-5D (Fig 1 green) and image classification tasks (Fig 2 green), we did not plot it in Fig 3 as it would be a repetition from Fig 2 and we wanted to avoid a graph too busy that is unreadable. We would love to know if there is an improvment we could make to let these existing experiments stand out on the first glance.
>
> **Current experiments cannot rule out that we simply observe a regularizer that prevents overfitting to the small dataset instead of transferring useful knowledge (the missing control experiment for this is Freeze RAND-polarity)**
>
> We would like to first point out that Freeze RAND-Polarity $\textbf{is included}$ in both our experiment on XOR-5D (Fig 1 green) and image classification tasks (Fig 2 green), we did not plot it in Fig 3 as it would be a repetition from Fig 2 and we wanted to avoid a graph too busy that is unreadable. We think both factors play a role in the performance gain we observe - by freezing polarities, we prevent overfitting; by pre-setting polarity pattern we transfer useful knowledge.
>
> **Theoreitcal derivation on capacity-speed tradeoff too simple**
>
> We include the exhaustive search result only for completeness and to support the reader's likely intuition. We are not making any claims about a specific learning rate improvement under SGD. It would be interesting to learn what actually makes the frozen polarity approach work better under SGD. This is an interesting future direction as we seek to better understand why we see this performance improvement.
>
> **Neuroscience claims overstated and vague**
>
> We think it's probably best for this paper to remove much of the discussion of neuroscience connections. There is simply too much detail one could rightfully go into that choosing almost any stopping point will be less than satisfying. We do not wish to get into the finer details of neural processing as the single bio-inspired element we wish to reflect in this work is the signed nature of excitatory and inhibitory synapses and its conceptual similarity to weight signs in an artificial neural network. We have removed and/or softened any wording that might suggest that we are claiming something about how or why the brain works the way it does.
>
> **Why citing Spitzer 2017?**
>
> "Neurotransmitter switching often appears to change the sign of the synapse from excitatory to inhibitory or from inhibitory to excitatory. In these cases, neurotransmitter switching and receptor matching thus change the polarity of the circuit in which they take place." Most neuroscientists were aware about neurotransmitter dynamic and switching $\textbf{within the same polarity}$ - the events where $\textbf{across polarity switch}$ are so rare that we cited this review paper in 2017 to encapsulate almost all of the behaviors that could observe such rare events. As stated in our paper introduction, such events were never discovered in the sensory cortices where image processing takes place in the brain (relevant to the task choice in the paper). This reference is to back our initial bio-inspiration claim that polarity switch indeed does not happen in adult animals, especially in the sensory cortices where image processing take place.

---

> > ### Author Response · Authors · 2022-11-19
> > **Contd**
> >
> > **Layer-by-layer polarity transfer**
> >
> > This is a very interesting future direction, but unfortunately not one we can pursue within the time constraints of this rebuttal period. One immediate thought is we have to understand how fixing first layer polarity affects the learning dynamic of the rest of the layers; as this will be crucial to draw any conclusion about whether configuring Gabor filters through polarity pattern is enough for learning fast.
> >
> > **Ideally repeat Fig. 3 with a validation set and early stopping instead of a fixed set of epochs, to rule out effects of overfitting on the very small datasets**
> >
> > The statistical efficiency results in Figure 3 are plotted for the validation error rate at convergence. We have edited our text to make this point more clear.
> >
> > **Question on citing Dale1935**
> >
> > To clarify, the bio-plausible paper we discussed immediately after the quoted sentence is from Cornford2021. We will remove the Dale1935 citation to avoid further confusion like such.
> >
> > **“As discussed in Sec 2, such an approach intrinsically put Frozen-Net in a disadvantageous position”. This is not entirely correct: the net is in a disadvantageous position in terms of representing any possible function - if the polarity is set correctly the net is in a highly advantageous position since the search-problem has been greatly reduced (under exhaustive search).**
> >
> > We've clarified that the disadvantage we're speaking of relates to how the current implementation may actually fight against the gradient in SGD. SGD learns through gradient update, and the frozen-polarity algorithm may fight back and refuse to make certain updates (changes). Theoretically this seems like it would slow down the search for convergence.

---

> > > ### Comment · Reviewer_8Q8m · 2022-11-28
> > > **Thanks for the clarifications.**
> > >
> > > Thank you for clarifying and amending the manuscript.

---

> > ### Comment · Reviewer_8Q8m · 2022-11-28
> > **Thank you for the detailed reply**
> >
> > I want to thank the authors for their detailed reply.
> >
> > **We are particularly interested in the small sample size cases because we this is the gap between natural and artificial intelligences we aim to close. To the best of our knowledge, we were not aware of an existing bio-plausible AI feed-forward DNN work that demonstrated more than matching the current state performance** Yes, the small sample regime is particularly interesting - from a pure transfer learning viewpoint (which I think is what the paper should focus on) it would be nice to see some SOTA results for that regime. The lack of empirical comparison with other methods cannot be justified by limiting the set of methods to **bio-plausible AI feed-forward DNN** (especially since the bio-plausibility is not given in the current work).
> >
> > **Missing control experiment Freeze RAND-polarity** Sorry for not being clear enough here. Freeze RAND as shown in Fig. 2 is by drawing polarities from Bernoulli(0.5). What's missing from Fig 3 is something like Freeze RAND, but using Bernoulli(\theta) where \theta is determined from the polarity ration of the trained ImageNet weights. As I wrote in my original comment, the point of this experiment is to see to which degree freezing random weights (with the right proportion of positive and negative weights) has a regularizing effect that explains the results. (In my original review I gave this the, admittedly clunky, name: "Freeze RAND-polarity experiment, where the correct proportion of polarities is frozen").
> >
> > **Neuroscience claims overstated and vague** Thanks for acknowledging this - however, given the current manuscript connections to neuroscience are still heavily implied throughout the manuscript (though the explicit claims in terms of explaining neuroscientific facts have been toned down). That leaves my improvements 1. and 2. mainly unaddressed.
> >
> > **NI design principle / bio-inspiration / bio-plausibility (in the abstract, intro, but also in the answer to my question: this reference is to back our initial bio-inspiration claim that polarity switch indeed does not happen in adult animals, especially in the sensory cortices where image processing take place.)** The new control experiment (fluid in polarity) shows that the observed effect is largely explained by good initialization rather than frozen polarity - which suggests that the *NI design principle* mentioned throughout the manuscript is largely irrelevant for artificial neural networks. This contradiction has not been addressed in the paper.

---

> > > ### Author Response · Authors · 2022-11-30
> > > **Thank you for your thoughts and clarification!**
> > >
> > > Now we understand your ratio-controlled RAND-Polarity freeze experiment idea. It is an interesting experiment to delineate which statistic/feature of polarity configuration is important to transfer knowledge through polarity. We will incorporate it into our final version!
> > >
> > > We would like to emphasize fixing the polarities **can** bring performance gain (Fig2 pink vs.~red, and see Fig3 for significance of pink). We recognize that in order to see the most performance gain, one should be smart about how they configure the polarities. This is why we also discussed the disadvantages of fixing polarities in Sec5 (Fig4). After all, natural intelligence not only fixes the polarities, but also benefits from the implicit knowledge embedded in the pre-configured polarity that transfers across generations.
> > >
> > > For your other two comments, please kindly find our view in the general comment on top. Thank you for being so engaged with us all along!

---

### Official Review · Reviewer_afAt · 2022-10-24

**Confidence:** 4
**Correctness:** 3
**Technical Novelty And Significance:** 3
**Empirical Novelty And Significance:** 2
**Recommendation:** 5

**Clarity, Quality, Novelty And Reproducibility:**

The writing needs more clarity. Please refer to the strengths and weaknesses section above for details.

**Strength And Weaknesses:**

**Strengths**

1. While I am not an expert in biological neural networks literature, it seems the fact that the weight signs don’t change throughout learning is an important observation and should be exploited more in the artificial neural networks.


**Weaknesses**

1. **Experiments**: I feel that the claims made by the authors, especially on ImageNet $\mapsto$ CIFAR10 transfer learning setup, are not well justified by the experiments.

* **Hyper-parameter setup**: The authors use a fixed learning rate 0.001 and number of epochs 100 for all the setups? How are these hyper-parameters determined? Could it be the case that this particular learning rate setting favors the fixed polarity baseline more?


* **Sec 3, ImageNet pre-training for polarity determination**: Obtaining weight polarities from the ImageNet trained model, and then training it on CIFAR10, and comparing it with randomly initialized model on CIFAR-10 seems unsatisfying. It is well-known in the over-parameterized network’s theory that the SGD trained network remains close to the initialization, so already by transferring the polarity from ImageNet to CIFAR-10, you start with a model with a good knowledge of the task making the overall comparison unfair.

* **Sec 4: Transferring polarity is better than transferring weights is not well-justified from the experiments**: It can be seen from Fig 3, that in fact transferring the weights and freezing them works better than transferring polarity in terms of statistical efficiency. Similarly, from A.3 it can be seen that for different thresholds of the validation accuracy different methods work better making it difficult to see whether there is a clear pattern where one method is consistently better. In fact, the authors themselves say that, given their experiments, “... polarity configuration is an effective medium, if not superior to weights pattern”. If it is not consistently superior I am not sure then what’s the point of the experiments and how can the authors conclude that polarity is a more effective and faster transfer mechanism. By the way, it could still very well be the case that in all these experiments the hyper-parameter settings are sub-optimal for the weight transfer baselines (as mentioned above).


* Overall, I feel that the authors only reported their experiments on two rather small settings, one of which was in fact a toyish synthetic setting. Even on CIFAR10, the experiments were not convincing enough to establish the points made by the authors.


2. **Writing**: It seems to me that the paper was written with a neuro-science audience in mind whereas most of the ICLR audience are ML researchers. Since the arguments made in the paper are not very difficult to follow so it may not make much of a difference. Nevertheless, it would be good to mold the paper along ML lines. For example, define the setup properly – metrics, train/ eval protocol etc, define what is a “Success rate”. Maybe even define what is the meaning of “polarity”. Overall, make paper more self-contained.


**Summary Of The Paper:**

The paper argues that in biological systems the weights’ polarities (signs) remain fixed, and only magnitudes (synaptic plasticity) change over time, leading to faster learning. Inspired by this argument, the authors propose to fix the polarity of weights in the DNNs to learn and transfer faster. Towards this the authors propose a learning algorithm, called Freeze-SGD, whereby given a predefined polarity of all the weights in the network, the weights’ signs are kept fixed throughout training and only magnitude is updated. The authors argue that proper task-based configuration of the polarity before training is important for faster learning. The authors further argue that transferring polarity instead of the weight patterns (magnitude + sign) is a more effective and compressed form of knowledge transfer. The experiments are conducted on classification tasks using a synthetic 5-XOR and image CIFAR-10 dataset.

**Summary Of The Review:**

Please refer to the strengths and weaknesses section above for details.

---

> ### Author Response · Authors · 2022-11-19
> **Interesting critics!**
>
> $\newcommand{\JBKK}{\textcolor{orange}{JBKK}}$
> $\newcommand{\REG}{\textcolor{green}{R5EG}}$
> $\newcommand{\afAt}{\textcolor{blue}{afAt}}$
> $\newcommand{\Qm}{\textcolor{violet}{8Q8m}}$
>
> We thank reviewer $\afAt$ for the interesting critics! We did one control experiment and many text editions to incorporate your concerns and suggestions, we will continue working on your hyper-parameter concern and incorporate them in the final version of this paper:
>
> **Hyper-parameter setup**
>
> $\textbf{lr}$: We controlled all of our scenarios using the same learning rate; we agree it is important to see if our result is robust across different learning rates. Due to the time constraints of this rebuttal period, we are not able to complete this experiment before the revision deadline; we will add these in the final version. $\textbf{epoch num}$: Our use of 100 epochs was selected on the basis of most models that ever converged doing so in 20-50 epochs.
>
> **Unfair comparison?**
>
> We believe the "unfair" comparison raised by reviewer $\afAt$ has been addressed by the control experiment raised by $\JBKK$, $\REG$, and $\Qm$. We have added the control experiments, please find them in Fig2-3 (we did not manage to get the XOR-5D done in time, but will add them in the final version). In short, we found 1) majority of the gain was brought by well-chosen polarity pattern (Fig 2). 2) fixing the polarity further improves on performance, albeit small. Such improvement is statistically significant (Fig 3 pink curve). Therefore our contribution is two folds: 1) we experimentally showed that polarity is an effective medium for embedding knowledge; 2) we experimentally showed fixing polarities helps networks learn faster. We point out for the second contribution, it could well be the case that by correcting the polarities during learning, we are resetting the learnt gradients every step and leading to much slower learning rate; quite on the contrary, even though freezing polarity is intrinsically disadvantageous when paired with SGD, we were still able to observe a performance gain, and such a gain is statistically significant across the 20 trials we ran. For a detailed statistical account, please kindly refer to the overall comment.
>
> **Transferring polarity is better than transferring weights is not well-justified from the experiments**
>
> Freeze IN-Weight works the best in terms of statistical efficiency, how could you conclude polarity transfer is superior to weight transfer? We disagree. First, Freeze IN-Weight does not perform better than Freeze IN-Polarity: with the additional magnitude information for weight transfer (Fig 3 cyan), there is nearly no gain in performance when data is limited (Fashion-MNIST $\leqslant 1000$, CIFAR-10 $\leqslant 1000$), and could even be detrimental to performance (e.g. CIFAR-10 500 and 1000 training samples). When we do observe a better performance from Freeze IN-Weight is when data is abundant, however, this is only when polarities are fixed; when polarities are fluid which is the traditional transfer learning setup (orange), it never outperforms Freeze IN-Polarity (red). We have edited our claims to emphasize that transferring polarities and fixing them is superior to transferring weights. Second, weight transfer consistently show wide performance variation in the CIFAR-10 dataset (appendix Fig B.3, cyan, orange) whereas polarity transfer (red, pink) delivers consistent good performance across trials.
>
> **Follow ML writing standards**
>
> We added a methodology section in the appendix that explicitly explained the metrics we used, training and evaluation procedures, success rate etc, in addition to the explanation we gave in the main text as terminologies arise. We are  happy to further improve and tailor our paper to the ML audience if the reviewer deem necessary.

---

### Official Review · Reviewer_R5EG · 2022-10-25

**Confidence:** 4
**Correctness:** 3
**Technical Novelty And Significance:** 3
**Empirical Novelty And Significance:** 3
**Recommendation:** 5

**Clarity, Quality, Novelty And Reproducibility:**

This work is of high quality and seems novel. Although the paper is generally well written, the authors could make it clearer. In particular, the methods and implementation details could be clearer and more organized. What it means for a function to be representable could be defined.



**Strength And Weaknesses:**

Strengths
- Interesting and important direction of research with potential improvements for AI and insights for both AI and neuroscience.
- If the brain is following this strategy it is a very interesting theory for why the brain has the restrictions it does.

Weaknesses
- The main contribution of this work is unclear to me. There seems to be two separate issues that are not adequately separated in the text and experiments: (1) the initialisation of a network with a task-specific pattern of signs that results in better learning (2) the freezing of signs during learning. The experiments lack the controls of “Fluid sufficient-Polarity”, and “Fluid IN-polarirty” which would be critical to understanding this. My suspicion is that (1) mainly underlies the results.
- The networks do not reflect the type of polarity restrictions as found in the brain (all of the outgoing connections from a neuron are of one polarity). Though the author’s acknowledge this in the discussion, it is not clear to me how this is work is a bridge.


**Summary Of The Paper:**

Connections between neurons in the brain are almost universally exclusively excitatory or inhibitory, which seemingly reduces the representational capacity of biological networks. Inspired by this observation, and the assumption that the brain has been optimized throughout evolution, the authors propose that by freezing the sign of connections between neurons may facilitate faster and more data-efficient learning.

**Summary Of The Review:**

While this is interesting and important work the main contribution is unclear to me - is freezing important or just choosing signs apriori. The connection to the biological networks seems lacking and should be developed in order to make the claim that this is the strategy the brain is employing.

---

> ### Author Response · Authors · 2022-11-19
> **Interesting critics!**
>
> $\newcommand{\JBKK}{\textcolor{orange}{JBKK}}$
> $\newcommand{\REG}{\textcolor{green}{R5EG}}$
> $\newcommand{\afAt}{\textcolor{blue}{afAt}}$
> $\newcommand{\Qm}{\textcolor{violet}{8Q8m}}$
>
> We thank reviewer $\REG$ for the excellent points! We did one control experiment and many text editions to incorporate your concerns and suggestions:
>
> **Fixing polarity vs. well-chosen initial polarity**
>
> We have added the control experiments, please find them in Fig2-3 (we did not manage to get the XOR-5D done in time, but will add them in the final version). In short, we found 1) majority of the gain was brought by well-chosen polarity pattern (Fig 2). 2) fixing the polarity further improves on performance, albeit small. Such improvement is statistically significant (Fig 3 pink curve). Therefore our contribution is two folds: 1) we experimentally showed that polarity is an effective medium for embedding knowledge; 2) we experimentally showed fixing polarities helps networks learn faster. We point out for the second contribution, it could well be the case that by correcting the polarities during learning, we are resetting the learnt gradients every step and leading to much slower learning rate; quite on the contrary, even though freezing polarity is intrinsically disadvantageous when paired with SGD, we were still able to observe a performance gain, and such a gain is statistically significant across the 20 trials we ran. For a detailed statistical account, please kindly refer to the overall comment.
>
> **How this work is a bridge to neuroscience?**
>
> Despite all of the differences from the brain mentioned here an in reviewer $\Qm$'s comments, by just adequately fixing the polarities, doing this one simple bio-inspired change, we could already **start** to close the gap between brains and DNNs on learning efficiency. These suggest polarity pattern is crucial, regardless of other systematic differences. We were excited about this first step and the initial evidence shown here that weight polarity is a direction worth exploring for both neuroscience and AI community, and even synergistically on the topic of polarity configuration. We recognize our work is far from a final answer in this direction thus removed our claims on neuroscience and only left it as an inspiration.
>
> **Methodology clarity**
>
> We added a methodology section in the appendix that consolidated all of our experimental details mentioned in the main text. The methodologies are not highlighted and explained as appropriate in the main text while still being accessible as a whole in the appendix.
>
> **What it means for a function to be representable?**
>
> We added an explanation towards the beginning of section 2 "With input space $\mathcal{X}$, a function $f$ is representable by a DNN $F$ when $\forall x \in \mathcal{X}, \epsilon>0, \; |f(x)-F(x)|<\epsilon$."

---

### Official Review · Reviewer_JBKK · 2022-10-25

**Confidence:** 3
**Correctness:** 4
**Technical Novelty And Significance:** 4
**Empirical Novelty And Significance:** 4
**Recommendation:** 8

**Clarity, Quality, Novelty And Reproducibility:**

The paper communicated its ideas very clearly and these ideas (to my knowledge) appear very novel. There is one point on weight initialisation that the authors should clarify for reproducibility and interpretability to ensure this paper has as much impact as it has the potential to have.

**Strength And Weaknesses:**

*Strengths:*
- A very interesting idea of broad and general interest to both the computational neuroscience community and the ML community.
- The authors explore the idea of fixed weight polarity is some depth, discussing the theoretical benefits and costs of fixed-polarity networks (costs: limits on representational capacity, pros: learning speed).
- Nicely designed experiments in both toy and more realistic settings (however see comment below on an important way they can be improved).
- Beautifully written.

*Weaknesses:*
- In all experiments, but most crucially in experiments 1 and 2: The authors should do the final control experiment in which they initialize the network with the same weights as Freeze-sufficient polarity (set to be the correct % polarity) but without the frozen learning algorithm. This will truly isolate the two factors of (1) initially setting the weights % with the correct polarity, and (2) keeping that weight polarity throughout training. This yields a fully factorized experimental design which is tidier for isolating the two contributing factors and their interactions. It may be that initializing the network with the correct polarities but without constraining these polarities via the update rule, is sufficient for the network to gain the same benefits. This experiment would tell us these sufficiency conditions.

- I couldnt see whether in each experiment the initial weights were set with the same generative process for the different experimental conditions (and so have the same magnitude distribution). To ensure that different weight magnitude distributions (which are known to affect learning rates) are not playing a role in these results, the Freeze- in polarity vs Freeze Rand, vs Fluid weights should all start with the same magnitude distribution, despite the differences in polarity. For these results to be interpretable this must be true - and so I assume this is the case - but I could not see this mentioned in the text (perhaps I missed it) and it’s quite an important feature to highlight.

- Could the authors comment on whether randomly initialised networks trained with SGD learn polarity early on in training? I am aware that weight magnitudes are typically learned very early on in training but am not sure whether polarity is also a commonly learned early feature of weights with SGD. If this is known, it would help to complete the picture and could be added to the discussion.


**Summary Of The Paper:**

This paper explores the computational features that result from a neural network with weights of fixed polarity. The authors demonstrate through proofs in constrained settings, and experiments in both constrained and more general settings, that networks with fixed and appropriately initialized weight polarity learn faster than randomly initialized networks. They also demonstrate that for randomly initialized networks of sufficient size, constraining the polarity of the weights in the update function does not compromise learning.

The paper proposes  a learning algorithm which is a variant on stochastic gradient descent but which does not permit polarity changes in the weights (Freeze-SGD). The paper compares networks trained under this algorithm (frozen nets) with networks trained under regular SGD, including networks that have previously had some pretraining. They present a theoretical trade-off between network representational capacity and learning speed, as a function of the network’s polarity being fixed.

In experiment 1 they consider single-layer networks on the 5D-XOR problem (the first 2 dims are classic XOR and the final 3 dims are noise). They compare learning rates of 3 networks to consider the influence of two factors: fixed weight polarities and well-chosen polarity patterns. The first net is initialised with the weights set to have the correct polarity combination for the task (a priori knowledge) and which uses a polarity-frozen update rule the authors propose. The second network uses this same update rule but is initialised with random small weights (presumably of the same scale), and the final network uses random weights and vanilla SGD. The network which had both the correct % polarity and used the polarity-preserving update rule learned the fastest.

In experiment 2 they consider image classification with AlexNet on the datasets Fashion-MNIST and CIFAR-10, and consider the same three initialisation and update conditions.

In experiment 3 they compare initialising the network with ImageNet-trained weights vs Imagenet-trained polarity %s, to assess whether transfer between tasks is more effective when transferring the weights magnitudes and polarities or simply the  polarity alone (both under the polarity-preserving update rule). This yielded a very surprising result that polarity alone resulted in faster learning on the new tasks.

The paper then calculates the probability that a randomly initialised network with a fixed polarity update rule will never be able to learn the XOR task. The experimental results closely match the theoretical results.


**Summary Of The Review:**

This paper explores the computational features that result from a neural network with weights of fixed polarity. The authors demonstrate through proofs in constrained settings, and experiments in both constrained and more general settings, that networks with fixed and appropriately initialized weight polarity learn faster than randomly initialized networks, and that for randomly initialized networks of sufficient size, constraining the polarity of the weights in the update function does not compromise learning. In my opinion the findings are novel and very interesting, but subject to a few caveats that need confirming.

---

> ### Author Response · Authors · 2022-11-19
> **Thank you for your constructive, insightful and positive feedback!**
>
> $\newcommand{\JBKK}{\textcolor{orange}{JBKK}}$
> $\newcommand{\REG}{\textcolor{green}{R5EG}}$
> $\newcommand{\afAt}{\textcolor{blue}{afAt}}$
> $\newcommand{\Qm}{\textcolor{violet}{8Q8m}}$
>
> We thank reviewer $\JBKK$ for your constructive, insightful and positive feedback! We did one control experiment and two analysis to answer your concerns and questions:
>
> **Fixing polarity vs. well-chosen initial polarity**
>
> We have added the control experiments, please find them in Fig2-3 (we did not manage to get the XOR-5D done in time, but will add them in the final version). In short, we found 1) majority of the gain was brought by well-chosen polarity pattern (Fig 2). 2) fixing the polarity further improves on performance, albeit small. Such improvement is statistically significant (Fig 3 pink curve). The added results give strong support on our claim that polarity is an effective medium for embedding knowledge; it gives support on our claim on fixing polarity helps learning faster (it is an interesting next step to adopt a learning algorithm fully compatible with fixed polarities to avoid the cost of fighting against the gradient in SGD). We have adjusted the claims in our paper accordingly to include these results.
>
> **Magnitude distribution at initialization the same across scenarios?**
>
> Yes! (except IN-Weight conditions as the transferred weights follow a different distribution than the randomly initialized magnitudes). This is quantified as a sanity check in appendix Fig B.6; and here is a description of the generative process: for all four conditions with RAND-Polarity or IN-Polarity, we randomly initialized the networks following normal procedures: Glorot Normal for conv layers, Glorot Uniform for fc layers; then either fixed the polarities as is (RAND-Polarity) or flipped the polarities according to IN template (IN-Polarity), by flipping, we introduced no change to the magnitude distribution.
>
> **Are polarities learnt early on?**
>
> We analyzed the number of polarity flips for Fluid RAND-Polarity for the first 50 epochs (see supp Fig B.5) and find polarities are mostly learned early on during training but still remain dynamic throughout the learning process (remain at an almost constant flipping counts once plateaued). It seems when training sample is small, the decay in polarity flip counts is slower, indicating more epochs are needed to reach plateau; however, we recognize such trend is confounded by smaller training set having less number of batches, therefore we did not draw conclusion on this point in the paper.

---

### Author Response · Authors · 2022-11-19
**We thank all reviewers for the constructive feedback!**

$\newcommand{\JBKK}{\textcolor{orange}{JBKK}}$
$\newcommand{\REG}{\textcolor{green}{R5EG}}$
$\newcommand{\afAt}{\textcolor{blue}{afAt}}$
$\newcommand{\Qm}{\textcolor{violet}{8Q8m}}$

We thank the reviewers for their insightful feedback! We are encouraged that they find our idea on weight polarity interesting ($\JBKK$, $\REG$, $\Qm$) and novel ($\JBKK$, $\REG$, $\Qm$), the direction important ($\JBKK$, $\REG$, $\afAt$) for both AI ($\JBKK$, $\REG$, $\afAt$) and neuroscience ($\JBKK$, $\REG$), the experiments nicely designed ($\JBKK$) and of high quality ($\REG$), and our writing clear ($\JBKK$, $\Qm$).

We highlight in this comment two themes we saw across the reviewers as a group.

**Freezing polarity vs. well-chosen initial polarity** @$\JBKK$, @$\REG$, @$\afAt$, @$\Qm$. There were two intermixed factors contributing to the performance gain of Freeze Sufficient-Polarity: 1) fixing polarity; 2) well-chosen polarity pattern. We added the control experiment Fluid Sufficient-Polarity (tested in Fashion-MNIST and CIFAR-10, see Fig 2-3 in paper). We found 1) majority of the gain was brought by well-chosen polarity pattern (Fig 2). For example, with 100 CIFAR-10 training samples, Freeze IN-Polarity gained $9.4\%$, Fluid IN-Polarity gained $8.4\%$, both compared to Fluid RAND-Polarity. The added results further support our claim that polarity is an effective medium for embedding knowledge, and often times superior to weight (polarity + magnitude) transfer (Fig 3); 2) fixing the polarity further improves on performance, albeit small. Such improvement can be statistically significant (Fig 3 pink curve, Fashion-MNISt $\geqslant 3000$ samples, CIFAR-10 $\geqslant 750$ samples). We point out here that it could well be the case that by correcting the polarities during learning, we are resetting the learnt gradients every step and leading to much slower learning rate; quite on the contrary, even though freezing polarity is intrinsically disadvantageous when paired with SGD, we were still able to observe a performance gain, and such a gain can be statistically significant across the 20 trials we ran. We will add the same control experiments to XOR-5D in the final version.

**Neuroscience claims** @$\REG$, @$\Qm$. We recognize that in our experiments, the networks do not obey Dale's rule, they have no recurrent connections (thus no dynamic), is not a spiking neuron model, and the SGD-based learning algorithm is not bio-plausible. Despite all of these differences from the brain, by just adequately fixing the polarities, doing this one simple bio-inspired change, we could already **start** to close the gap between brains and DNNs on learning efficiency. These suggest polarity pattern is crucial, regardless of other systematic differences. We recognize this is not the final answer therefore we have taken out all of our neuroscience claims and only kept it as a source of inspiration; we are, however, excited about this first step and the initial evidence shown here that weight polarity is a direction worth exploring for both neuroscience and AI community.

---

> ### Comment · Reviewer_JBKK · 2022-11-20
> **Thanks for the response and edits**
>
> I want to thank the authors for their responses and the other reviewers for their in-depth reading.
> I am not really an expert on this topic, but I do find the observation that weight polarity is a critical feature of neural net learning to be a very interesting one, if it is supported by the data. I personally dont mind the hand-wavey links to neuroscience (as a neuroscientist myself), as features of learning that generalise between biological and artificial NNs can exist at various different levels of abstraction and still be useful for shaping the thinking in each field.
> I would like to see this paper published, as I think many others will similarly find this work interesting and so I will keep my score as is. However if the consensus amongst the other reviewers and AC is a reject, I would encourage the authors to disseminate this work on arXiv anyway so that it is available to the community to read.

---

### Author Response · Authors · 2022-11-30
**Sum of our contribution**

$\newcommand{\JBKK}{\textcolor{orange}{JBKK}}$
$\newcommand{\REG}{\textcolor{green}{R5EG}}$
$\newcommand{\afAt}{\textcolor{blue}{afAt}}$
$\newcommand{\Qm}{\textcolor{violet}{8Q8m}}$

We agree with $\JBKK$~that the connection between ``biological and artificial NNs can exist at various different levels of abstraction and still be useful for shaping the thinking in each field.'' We also recognize there are different views on this matter and evaluating which level is the right one will be subjective. To not be side-tracked by this still debatable topic in the field and keep the focus on our main contribution, we decided to take out all neuroscience conclusions in the paper (see revision) but kept our bio-inspiration observations.

**Our main contribution is showing that weight polarity is a critical feature for a network to learn fast, with less data.** There are two contributing factors to the performance gain: 1) fixing the polarity throughout training (Fig2 red vs. pink; statistical significance in Fig3 pink); 2) informatively configure the polarity (polarity transfer, Fig2 red vs. green, also pink vs.blue).

---

### Decision · Program_Chairs · 2023-01-20

**Decision:**

Reject

**Justification For Why Not Higher Score:**

Reviewers and myself were ultimately able to reach a consensus based on reevaluating the contributions and updated paper/rebuttals. It was decided that although it's been much improved from previous versions, it would benefit from a significant reframing toward being more about transfer learning rather than neuroscience, and more extended experiments / controls along these lines.

**Justification For Why Not Lower Score:**

N/A

**Metareview: Summary, Strengths And Weaknesses:**

As this was a borderline paper, with wildly diverging scores (ranging from 3 to 8), a discussion was arranged and attended by all reviewers and myself. Reviewers were asked to summarize the main points of their reviews and consider whether they wanted to maintain their scores in light of seeing other reviews and authors’ rebuttal.

This paper proposes training neural networks using constrained polarity on the weights, inspired by ideas from neuroscience, and suggests that good initialization of this polarity can contribute to better transfer learning. There was initially a divergence of opinion on this paper, but after an extended discussion, all reviewers agreed that it’s not ready for publication in its current form. This decision was based on the following considerations:
1. Although the ties to neuroscience are interesting, they are mainly superficial and without justification. The findings don’t really have implications for neuroscience as neural connections in the brain are constrained in many other ways (e.g. Dale’s law). Reviewers felt that the framing of the contributions in these terms was therefore unjustified, and therefore the main contributions were on the transfer learning / ML side.
2. The additional control experiment that was added during the rebuttal period showed that much of the transfer learning benefit was due to the fact that polarities were initialized using a priori information, rather than that the polarities were constrained during further training. Freezing the polarities did confer a small added significant benefit, but an additional control experiment was missing to show that this wasn’t due to the added regularization effect of having frozen polarities (suggested by 8Q8m).
3. This effect is quite small and it’s not clear if this is practically useful in any real ML setting, beyond toy tasks.

Although I can’t recommend acceptance at this time, I note that reviewers overall thought it was an interesting paper and worthy of being pursued further. They have suggested several other experiments that would really help to bolster the claims made and flesh out the contributions, and I hope that these are helpful for future iterations of this work.


**Summary Of Ac-Reviewer Meeting:**

Notes from meeting held on Dec 6, 2022

R5EG: 2 directions it’s interesting
* Why might brain use these constraints
* How might these be implemented
* Additional experiment is as expected
* Biological side is still interesting but paper has overemphasized this

JBKK
* Nice idea that, if well justified, could really influence the way that people think
* Going through other reviews, there might be reason to question that
* Agrees with other reviewers that links to neuroscience need to be more firm
* TLDR: is happy for mind to be changed on this, doesn’t ultimately feel that strongly

8Q8m
* Not convinced by connection to neuroscience, is mainly a superficial connection
* So this leaves the contribution being only regarding transfer learning
* But a large part of the results seems to depend on fortuitous initialization of polarity, although a significant benefit still remains
* The additional control experiment needed to be done was not done (though there was a misunderstanding of the experiment that the reviewer asked for in their original review)
* Overall is interesting but still needs to be fleshed out
* Still too much emphasis on neuroscience links

R5EG
* Intuitively, freezing polarity should not work
* Main effect happens at small sample sizes

afAt
* Approached from ML perspective
* Main concern was with model selection, perhaps the model they selected was optimal for freezing polarity, so not fair to compare to not freezing. Need to do some validation
* Author response was that they didn’t have enough time.

R5EG: This was during SFN and also cosyne abstract deadline, so understandable they might’ve run out of time